# Myeloid transformation by *MLL-ENL* depends strictly on C/EBP

Radoslaw Wesolowski[1,*], Elisabeth Kowenz-Leutz[1,*], Karin Zimmermann[1,†], Dorothea Dörr[1,†], Maria Hofstätter[1,†], Robert K Slany[2], Alexander Mildner[1] , Achim Leutz[1,3]

**Chromosomal rearrangements of the mixed-lineage leukemia gene *MLL1* are the hallmark of infant acute leukemia. The granulocyte-macrophage progenitor state forms the epigenetic basis for myelomonocytic leukemia stemness and transformation by MLL-type oncoproteins. Previously, it was shown that the establishment of murine myelomonocytic *MLL-ENL* transformation, but not its maintenance, depends on the transcription factor C/EBPα, suggesting an epigenetic hit-and-run mechanism of MLL-driven oncogenesis. Here, we demonstrate that compound deletion of *Cebpa/Cebpb* almost entirely abrogated the growth and survival of *MLL-ENL*–transformed cells. Rare, slow-growing, and apoptosis-prone *MLL-ENL*–transformed escapees were recovered from compound *Cebpa/Cebpb* deletions. The escapees were uniformly characterized by high expression of the resident *Cebpe* gene, suggesting inferior functional compensation of C/EBPα/C/EBPβ deficiency by C/EBPε. Complementation was augmented by ectopic C/EBPβ expression and downstream activation of IGF1 that enhanced growth. *Cebpe* gene inactivation was accomplished only in the presence of complementing C/EBPβ, but not in its absence, confirming the *Cebpe* dependency of the *Cebpa/Cebpb* double knockouts. Our data show that *MLL*-transformed myeloid cells are dependent on C/EBPs during the initiation and maintenance of transformation.**

## Introduction

Mixed-lineage leukemia (MLL) represents an aggressive pediatric cancer of the blood, with features of acute lymphoblastic leukemia and acute myeloid leukemia (AML). Chromosomal translocations at 11q23 are predominant in MLL and fuse the N-terminal part of the Trithorax-like MLL1/KMT2A methyltransferase to multiple partner proteins (Shilatifard, 2012; Slany, 2016). *MLL*-induced AML may originate from hematopoietic stem cells (HSCs) and/or an early

progenitor state and involve the establishment of leukemic stem cells (LSCs) that maintain lineage plasticity and an intermediate lymphoid–myeloid immunophenotype (Daigle et al, 2011; Goardon et al, 2011; Chen et al, 2013; Krivtsov et al, 2013). Despite recent advances in understanding the molecular mechanism of the disease, therapy of *MLL* translocation-induced leukemia remains a clinical challenge.

The prevalent leukemic *MLL* translocations entail genes encoding components of the super elongation complex, including *ENL*, *AF9*, and *AF4*. Both MLL-ENL and MLL-AF4 represent potent fusion oncoproteins that experimentally transform murine bone marrow cells in vitro (Smith et al, 2011). Mechanistically, MLL fusion oncoproteins stimulate the expression of target genes, including critical genes of the HOXA cluster, by co-recruiting the DOT1L complex and by promoting DNA polymerase II pause release and the elongation phase of gene transcription (Okada et al, 2005; Krivtsov & Armstrong, 2007; Krivtsov et al, 2013). Deregulated expression of the MLL target genes *Hoxa9* and *Meis1* partially recapitulate leukemogenic self-renewal and eventually cause experimental leukemogenesis (Collins & Hess, 2016b).

C/EBP (CCAAT enhancer–binding protein) family members are transcription factors that may function as activators and repressors depending on the cellular and molecular context and the expression status of the C/EBP protein isoforms (Zahnow, 2002; Nerlov, 2004; Johnson, 2005). C/EBPα, a master regulator of granulocyte-macrophage progenitor (GMP) biology, is also of central importance to leukemic myelomonocytic transformation. C/EBPα controls the transition from common myeloid progenitors to GMPs and prevents exhaustion of the HSC compartment (Zhang et al, 2004). C/EBPα-deficient progenitors resist transformation by *MLL-ENL*, *MLL-AF9*, *MOZ-TIF2*, and *Hoxa9/Meis1*. Interestingly, after the establishment of *MLL-ENL* transformation, C/EBPα can be removed genetically, whereas the malignant phenotype persists (Ohlsson et al, 2014). These findings suggest that, in the presence of C/EBPα, a "hit-and-run"–type *MLL* transformation consolidates an epigenetic state that is maintained in the absence of the initial

[1]Max Delbrück Center for Molecular Medicine, Berlin, Germany [2]Department of Genetics, Friedrich-Alexander University Erlangen-Nürnberg, Erlangen, Germany [3]Institute of Biology, Humboldt University of Berlin, Berlin, Germany

Correspondence: aleutz@mdc-berlin.de
*Radoslaw Wesolowski and Elisabeth Kowenz-Leutz contributed equally to this work
†Karin Zimmermann, Dorothea Dörr, and Maria Hofstätter contributed equally to this work

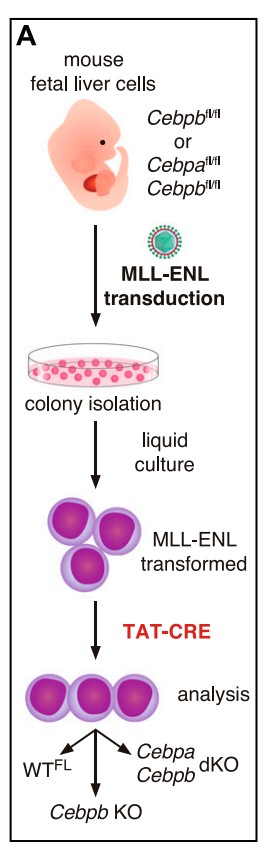

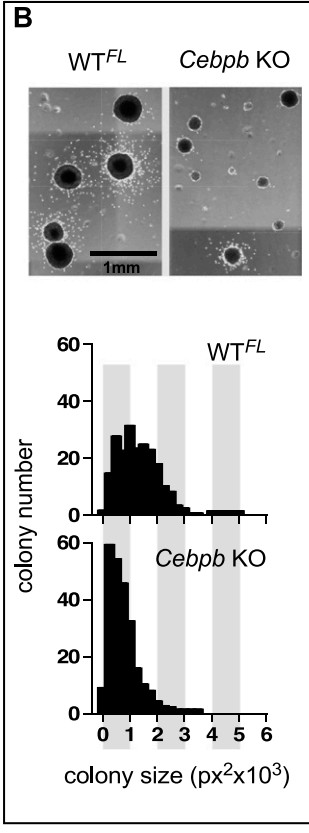

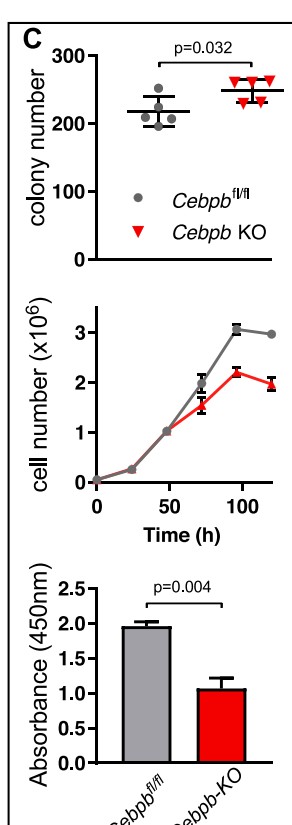

**Figure 1.** *Cebpb* Deletion in *MLL-ENL*–Transformed Cells.
**(A)** Schematic illustration of murine tissue culture *MLL-ENL* leukemia model. Top to bottom: Murine fetal liver cells from WT^FL animals were transduced with *MLL-ENL* and selected with G418 for 14 d in liquid culture. Subsequently, the cells were seeded in semi-solid methylcellulose medium. Single colonies were isolated, expanded in liquid culture, and assessed for *MLL-ENL* integration. Next, the cells were treated with TAT-Cre recombinase to remove floxed *Cebp* alleles. Gene excision was determined by PCR (see Fig S1B). **(B)** Top: representative microscopic scans of semi-solid methylcellulose cultures with WT^FL and *Cebpb* KO colonies. Cells were seeded at a density of 5,000 cells per 35-mm well, and colonies were scored after 10 d. Bottom: colony size distribution (see also Fig S1C). **(C)** Top: WT^FL and *Cebpb* KO colonies in semi-solid methylcellulose medium. Middle: Growth curves of WT^FL and *Cebpb* KO colonies. Bottom: WST-1 assay showing the effect of *Cebpb* removal on *MLL-ENL*–transformed cells. Values are the mean ± SD (two-tailed Mann–Whitney U test, **$P$ < 0.005).

inducing transcription factor C/EBPα (Roe & Vakoc, 2014). Alternatively, other C/EBP family members that have not been examined or that have remained undetected may be involved in maintaining the myelomonocytic and transformed state.

Here, we considered the role of C/EBPβ in maintaining the myeloproliferative *MLL-ENL*–transformed state. Using a somatic genetics approach, we show that the viability and proliferation of *MLL-ENL*–transformed mouse myeloblasts were impaired by removing C/EBPβ and were almost entirely abrogated by compound deletion of the *Cebpa* and *Cebpb* genes. Strikingly, all of the few surviving *Cebpa*/*Cebpb*–deficient *MLL-ENL*–transformed clones consistently expressed C/EBPε. Our data suggest that both the initiation and the maintenance of *MLL-ENL* transformation depend on transcription factors of the C/EBP family and imply a therapeutic opportunity in interfering with the C/EBP dependency of *MLL* transformation.

# Results

### Removing C/EBPβ slows the growth of *MLL-ENL*–transformed cells

Experimentally induced murine leukemogenesis and acquisition of the LSC state by the *MLL-ENL*, *MLL-AF9*, and *MOZ-TIF2* oncogenes depend on C/EBPα that induces myeloid commitment, or in the absence of C/EBPα, on the alternative establishment of the GMP state (Ye et al, 2015). This suggests that establishment of the GMP phenotype is sufficient and subsequently epigenetically memorized, in concordance with transformation by *MLL-ENL*. However, the alternative induction of the GMP state in the absence of C/EBPα requires C/EBPβ (Hirai et al, 2006; Manz & Boettcher, 2014). This raises the possibility that C/EBPβ may be involved in maintaining the LSC state and the leukemic process.

To distinguish between these possibilities, murine fetal liver cells derived from conditional *Cebpb*^fl/fl or *Cebpa*^fl/fl/*Cebpb*^fl/fl (designated WT^FL) mice were retrovirally transformed by the *MLL-ENL* oncogene, as outlined in Fig 1A. Briefly, *MLL-ENL*–transduced liver cells were seeded in methylcellulose medium containing cytokines (IL-3, IL-6, and stem cell factor [SCF]), and the emerging transformed colonies were isolated after 10–14 d and expanded in cytokine-supplemented liquid medium. *MLL-ENL* transformation was evident by the stable growth of immortalized cells that could be propagated in the presence of IL-3. Cytofluorometric analysis showed that the majority of cells expressed myeloid lineage surface antigens (CD16/32 and CD11b), and that >25% of the transformed populations also expressed c-Kit, a marker of HSC and early progenitor cells (Fig S1A), suggesting a progenitor/GMP-type phenotype, in agreement with published data (Somervaille & Cleary, 2006). After myeloid cell transformation had been completed, MLL-ENL clones were treated with cell membrane-permeable recombinant TAT-Cre (Brown & Byersdorfer, 2017) to determine the biological effect of *Cebpb* gene deletion on transformation (Fig S1B). Fig 1B and C show that removing *Cebpb* from the *MLL-ENL*–transformed progenitors had

little effect on total colony numbers; however, the secondary colonies derived from *MLL-ENL* C/EBPβ KO cells were smaller than the colonies derived from *MLL-ENL* cells with an intact *Cebpb* allele (Figs 1B and S1C). Cells derived from *Cebpb* KO cells showed decreased growth rates and diminished metabolism and viability in culture medium containing IL-3 (Fig 1C, cell counts and WST-1 to formazan conversion) as compared with the wild-type WT^FL *MLL-ENL*–transformed cells, in accordance with attenuated growth in semi-solid and liquid cultures. We conclude that, although *Cebpb* is not essential for the clonogenicity of *MLL-ENL*–transformed cells, it enhances their proliferation and viability.

## Simultaneous deletion of *Cebpa/Cebpb* inhibits *MLL-ENL*–transformed cell proliferation

To generate compound *Cebpa/Cebpb* KO cells, two parental *Cebpa*^fl/fl/*Cebpb*^fl/fl *MLL-ENL* clones were infected with a retrovirus encoding a conditional Cre recombinase, and *Cebp* allele deletion was monitored by PCR. Cre-mediated gene deletion was readily observed; however, even after prolonged Cre recombinase activation over 5 d, we failed to delete all four *Cebp* alleles over numerous attempts (data not shown). In an alternative approach, four parental *Cebpa*^fl/fl/*Cebpb*^fl/fl MLL-ENL clones were treated with recombinant TAT-Cre, which also resulted in allelic Cebp heterozygosity (Brown & Byersdorfer, 2017). Clones with partial *Cebp* deletions were then pooled, expanded in mass culture (to ~4 × 10^6 cells), and re-treated with cell-permeable TAT-Cre recombinase to force deletion of the remaining *Cebpa/Cebpb* alleles. The final round of Cre treatment resulted in a widespread crisis of cell growth and cell death. All surviving cells were directly seeded in semi-solid medium for scoring potential *Cebpa/Cebpb* double KO (dKO) on a clonal basis. In total, 1,056 colonies (from ~1,200 to 1,300 colonies) were isolated and examined for *Cebpa/Cebpb* deletion. Only 14 *Cebpa/Cebpb* dKO subclones from two mice (2 and 12 subclones, respectively) could be identified, suggesting that the transformed *MLL-ENL* cells were strongly dependent on *Cebpa/Cebpb* gene expression.

We next performed phenotypical analysis of WT^FL, *Cebpb* KO, and *Cebpa/Cebpb* dKO cells by flow cytometry (Fig 2). We detected no differences in the frequencies of CD11b+ or Ly6C+ cells between the WT^FL and *Cebpb* KO cells, suggesting that the deletion of *Cebpb* was compatible with maintaining the progenitor phenotype. *Cebpb* KO cells, however, showed diminished development of mature Ly6G+ neutrophils, whereas differentiation into CD115+ monocytes or macrophages in vitro was unaffected (Fig 2A). In contrast, compound deletion of *Cebpa* and *Cebpb* led to reduced CD11b and Ly6C reactivity, indicating a more immature or neomorphic phenotype (Fig 2A). Accordingly, neither Ly6G+ neutrophils nor CD115+ monocytes or macrophages could be derived from the *Cebpa/Cebpb* dKO cells (Fig 2A). Analysis of the early and late apoptotic stages in the WT^FL, *Cebpb* KO, and Cebpa/Cebpb dKO clones showed that deleting *Cebpb* and *Cebpa/Cebpb* partially and severely enhanced the apoptosis rate, respectively (Fig 2A, right). Next, we analyzed the cell proliferation rate by fluorescent CFSE dye dilution using flow cytometry. As shown in Fig 2B, cells from all genotypes were characterized by high proliferation rates. However, both single *Cebpb* and compound

*Cebpa/Cebpb* dKO cells showed reduced proliferation as compared with the WT^FL cells.

Histological staining of cytospins of WT^FL, *Cebpb* KO, and *Cebpa/Cebpb* dKO cells are shown in Fig 2C. WT^FL cells characteristically exhibited a predominant monoblastic/myeloblastic appearance, sometimes with kidney-shaped nuclei, relatively pale cytoplasm, and few, mostly diffuse cytoplasmatic granules. *Cebpb* KO cells displayed a similar cytoplasm/nucleus ratio to WT^FL cells, with early monocytic/myelocytic appearance, some vacuoles, and most typically, azurophilic granules in the cytoplasm. The *Cebpa/Cebpb* dKO cells had a smaller cytoplasm/nucleus ratio, with more darkly stained cytoplasm and frequently hyposegmented nuclei. These data suggest that the different genotypes are also reflected in distinct early myelomonocytic phenotypes.

Comparative serial replating of the WT^FL, *Cebpb* KO, and *Cebpa/Cebpb* dKO cells revealed a steady increase in the clonogenicity of the WT^FL and *Cebpb* KO cells and a decline in that of the *Cebpa/Cebpb* dKO cells (Fig 2D), suggesting enrichment for transformed stem cells in the WT^FL and *Cebpb* KO cells and the loss of stemness in the *Cebpa/Cebpb* dKO cells during replating. The characteristic heterogeneous appearance of the compact and disperse WT^FL colonies and more uniform, round, and compact *Cebpb* KO colonies was maintained during replating, whereas *Cebpa/Cebpb* dKO colonies continuously declined during replating and ceased growth beyond the fourth replating (Fig 2E). However, differences in the colony-forming capacity between the WT^FL and *Cebpb* KO cells became evident after partial withdrawal of cytokines (Fig 2F, growth in IL-3 medium), revealing that the *Cebpb* KO cells were more dependent on the standard cytokine mix (IL-3, SCF, IL-6, and GM-CSF) than the WT^FL cells.

## Endogenous C/EBPε compensates for C/EBPα and C/EBPβ deficiency in *MLL-ENL*–transformed cells

Related gene products may functionally compensate for distinct deleted genes, and obscure otherwise severe phenotypes. In particular, this can be observed in gene families that share evolutionarily conserved origins, such as the C/EBP family (El-Brolosy & Stainier, 2017). Therefore, we wondered about the compensatory mechanisms that may have occurred in the *Cebpa/Cebpb* dKO *MLL-ENL* clones to permit their survival. RNA sequencing (RNA-seq) analysis of two randomly chosen dKO clones derived from different animals confirmed the absence of *Cebpa* and *Cebpb* expression (Fig 3A). Strikingly, both dKO *MLL-ENL*–transformed clones showed up-regulated *Cebpe* gene expression, whereas *Cebpd* expression remained largely unchanged in the WT^FL and dKO cells. These data suggest that the activation of *Cebpe* may compensate for the loss of *Cebpa/Cebpb*. Accordingly, we examined all 14 recovered dKO clones for expression of the C/EBP family members by protein blotting. Fig 3B shows that all dKO clones strongly expressed the C/EBPε protein, whereas only small and inconsistent changes were observed in C/EBPδ, C/EBPγ, and C/EBPζ protein levels. These data are in line with previous findings showing that ectopic expression of C/EBPε, similarly to C/EBPβ but unlike C/EBPα, is compatible with the proliferation of GMP-like progenitors (Cirovic et al, 2017).

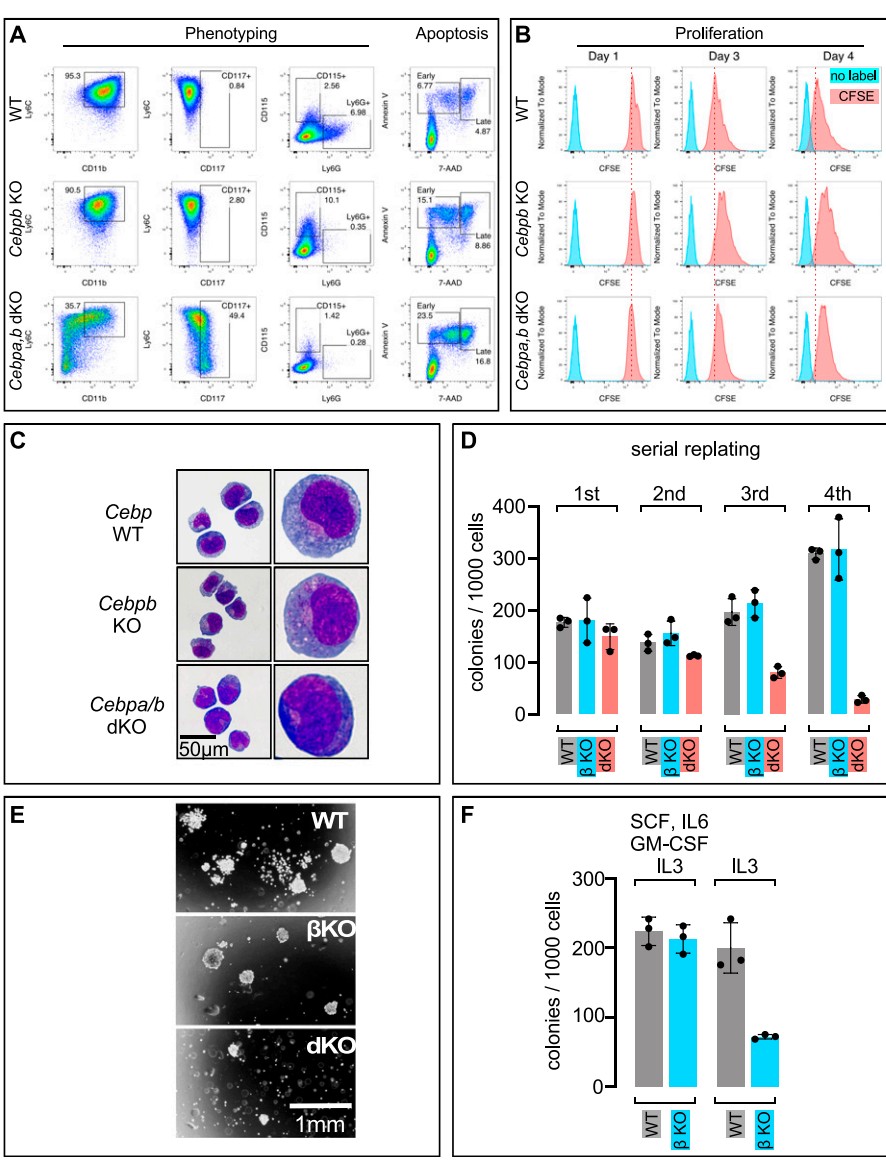

**Figure 2.** Characterization and comparison of *MLL-ENL*–transformed WT[FL], C/EBPβ KO, and C/EBPα/C/EBPβ dKO cells.

**(A)** Flow cytometric phenotyping of WT[FL], *Cebpb* KO, and *Cebpa/Cebpb* dKO cells using CD11b, Ly6C, CD117, CD115, and Ly6G markers (left panel). Apoptosis was examined via annexin V staining in combination with 7-AAD (right panel). Two independent experiments were performed, yielding similar results. **(B)** Analysis of frequency of cell division using CFSE in flow cytometry. Cell types as in A and cells were analyzed 1, 3, and 4 d after CFSE loading. Two independent experiments were performed, yielding similar results. **(C)** Cytospins were prepared from exponentially growing cell types and stained with Giemsa/eosin. Micrographs were taken from small groups of cells (left) to show uniformity and from enlarged single cells (right) to show differences in subcellular features. **(D)** Serial replating in methylcellulose/Iscove's DMEM supplemented with IL-3, stem cell factor, GM-CSF, and IL-6. Cells were seeded in triplicate, grown for 7 d, and colonies were counted before reseeding at 5,000 cells/well. Two independent experiments and for third and fourth replating three experiments were performed, yielding similar results. **(E)** Micrographs of colonies from fourth replating. **(F)** Cells from the third replating round were seeded in semi-solid medium supplemented with the complete cytokine cocktail or IL-3 only, as indicated. Colonies were counted after 7 d. Three independent experiments were performed, yielding similar results.

All 14 recovered *Cebpa/Cebpb* dKO clones grew slowly in liquid culture and had increased apoptosis, as compared with the WT[FL] *MLL-ENL*–transformed cells (compare with Fig 2A, right), suggesting incomplete functional compensation by C/EBPε. To determine whether distinct C/EBP isoforms could rescue *MLL-ENL*–transformed cell growth, isoforms of C/EBPα (p42 and p30) or C/EBPβ (LAP*, LAP, and LIP) were retrovirally transferred and subsequently enriched for co-expression of GFP, as shown in Fig 4A. As expected, growing cells that expressed the proliferation-suppressive C/EBPα p42 isoform could not be recovered, and the few GFP+ sorted cells that grew out failed to express the p42 protein, confirming successful retroviral transduction but failure of genetic complementation by C/EBPα p42. Growing cells were obtained for all other constructs (Fig 4A), and their proliferation rates, clonogenicity, and viability were assessed. Colony formation of the parental dKO cells was partially rescued by the C/EBPα p30 isoform (Fig 4B–D), and to a higher extent by the C/EBPβ LAP* isoform,

whereas the C/EBPβ LAP and C/EBPβ LIP isoforms displayed intermediate and no discernible complementation capacity, respectively (Figs 4E–G and S2). In summary, the C/EBPβ LAP* isoform, and in part, the C/EBPβ LAP or C/EBPα p30 isoforms, restored dKO *MLL-ENL*–transformed cell proliferation (Fig 4B and E) and viability (Fig 4C and F).

Next, we examined the role of C/EBPε in *Cebpa/Cebpb* dKO *MLL-ENL* cells that expressed high levels of endogenous C/EBPε, termed C/EBPε+, by RNA interference strategies. However, the fragile dKO *MLL-ENL* C/EBPε+ cells did not tolerate the small interfering RNA treatment, and various conditionally inducible, retrovirally delivered short hairpin RNAs failed to down-regulate C/EBPε reproducibly (data not shown). The data may suggest that triple deletion of the *Cebpa*, *Cebpb*, and *Cebpe* genes is incompatible with cell survival. Interestingly, we observed that expression of the endogenous C/EBPε ceased after extended cultivation of *MLL-ENL* dKO cells that were complemented with the C/EBPβ isoforms LAP*,

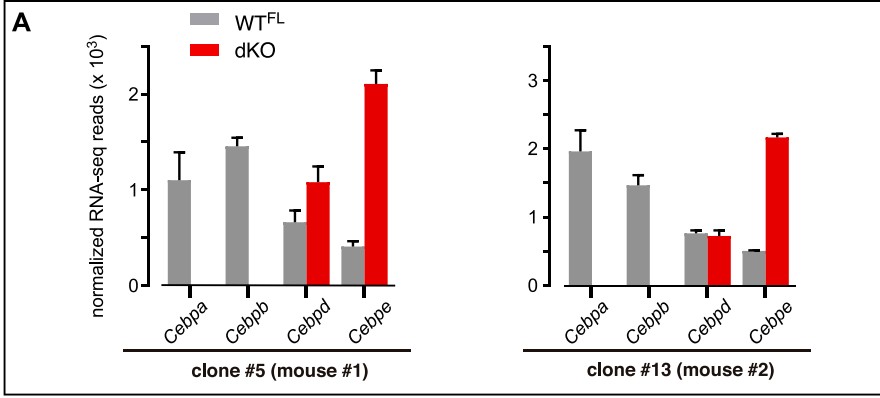

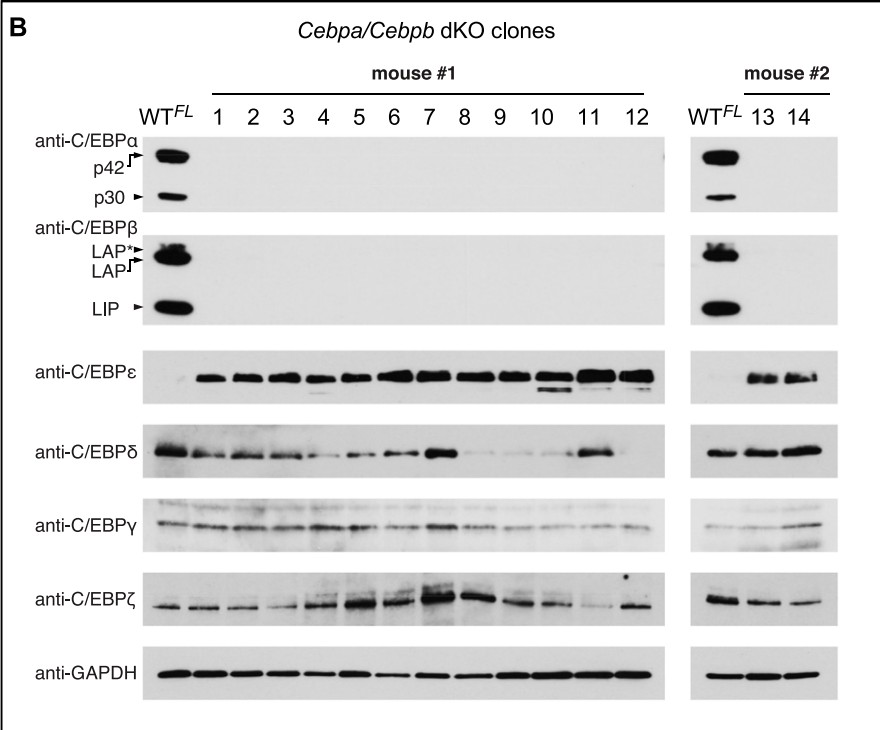

**Figure 3.  *MLL-ENL*–Transformed dKO Clones Express C/EBPε.**
**(A)** Data showing normalized RNA-seq read counts for indicated genes of the *Cebp* family in WT[FL] (gray) and two independent *Cebpa/Cebpb* dKO *MLL-ENL*–transformed cell clones (red) derived from different mice. RNA-seq was performed in triplicates. **(B)** Immunoblots show the expression of resident C/EBP family proteins from WT[FL] and dKO transformed *MLL-ENL* cells. The panel summarizes C/EBP family protein expression in all 14 dKO clones from two different mice (clone #1–12, mouse 1; clone #13 and #14, mouse 2).

and to a lesser extent with LAP, but not with LIP (Fig 5A). Based on the observation of an inverse correlation between resident C/EBPε[+] protein levels before and after ectopic C/EBPβ LAP* expression (Fig 5A), we explored the mutability of *Cebpe* in dKO *MLL-ENL* cells in the presence or absence of ectopically expressed C/EBPβ LAP*, as outlined in Fig 5B. Briefly, targeted deletion of resident *Cebpe* by Cas9 was examined after transduction of dKO cells with either empty control or C/EBPβ LAP* retrovirus in parallel, both expressing GFP as a marker. GFP[+] cells from both approaches were sorted, expanded, and infected with the same batch of a genome editing vector, encoding blue fluorescent protein (BFP) as a marker and Cas9 plus guide RNAs (Li et al, 2016; Henriksson et al, 2019), targeting *Cebpe* exon 1. After infection, sorted BFP[+] cells were expanded, seeded in semi-solid medium, and individual colonies were isolated from both vector control and C/EBPβ LAP* expressing cells (Fig 5B).

In comparison to vector control cultures that showed many abortive small colonies, augmented colony formation and proliferation were already discernible at an early stage during colony formation in C/EBPβ LAP*-complemented dKO cells (data not shown). After single colonies had been isolated, proliferation in liquid culture was discontinued in 35.9% of the vector control clones (N = 79/220 clones), but only 17.8% (N = 32/180 clones) of the C/EBPβ LAP* supplemented dKO clones were abortive (Fig 5C). Only 12.3% (27/220) of the vector controls, yet 48.9% (88/180) of the C/EBPβ-LAP* dKO clones showed robust growth in liquid culture. Among the vector control clones, all 27 properly growing control clone isolates and 31 (out of 88) randomly chosen C/EBPβ LAP*-complemented clones were expanded in liquid culture. As shown by protein blotting (Fig 5D), each of the five LAP*-dKO and five vector controls showed reciprocal expression of LAP* and C/EBPε. Analysis of the genomic *Cebpe* status (Fig 5E) showed that 80% (16/27 clones)

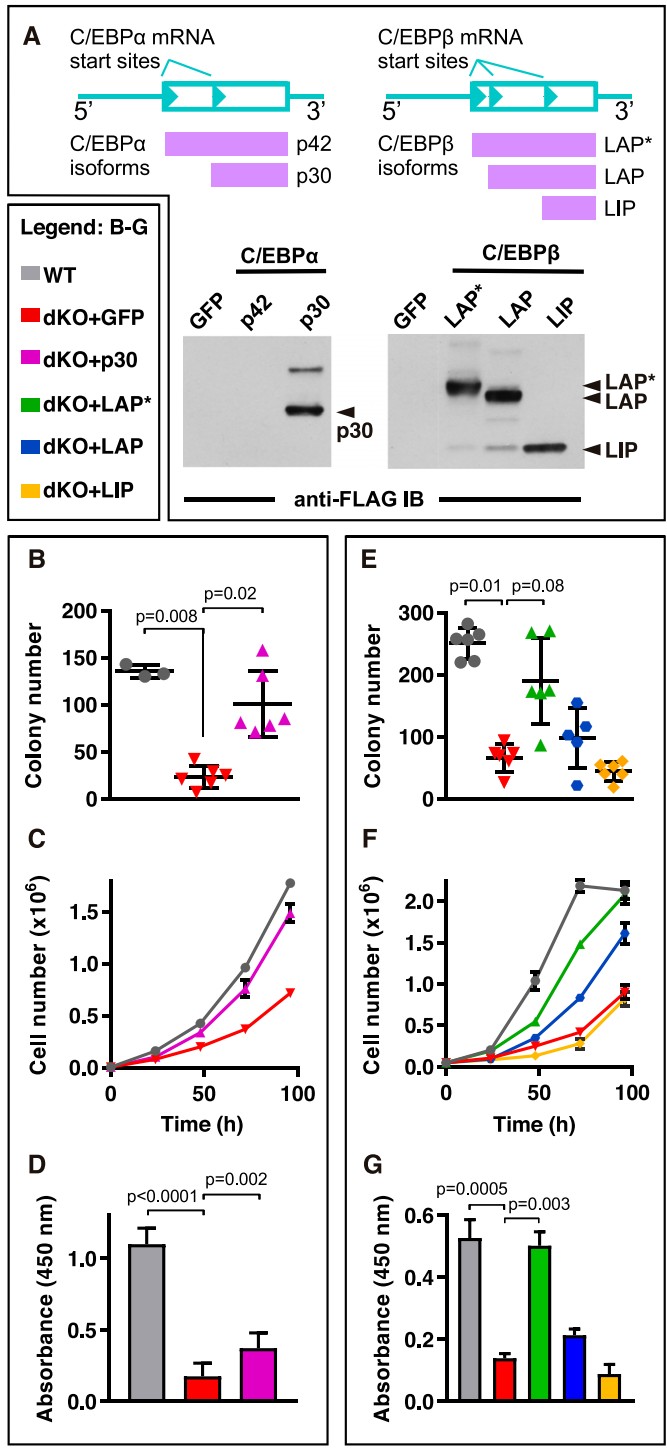

of the vector control clones retained the WT *Cebpe* genotype, and only two showed indel frameshift mutations (see below). In contrast, 74% (20/31 clones) of the C/EBPβ LAP*–complemented clones showed biallelic frameshift indels and/or large deletions. Importantly, 7/27 vector control clones but only 2/31 of LAP* clones had in-frame mutations. Both C/EBPβ LAP*–complemented in-frame mutations were associated with frameshift indels in the second *Cebpe* allele, whereas both frameshift indels of the vector control cells retained either a WT or an in-frame deletion in the second allele, respectively. Most remarkably, all in-frame mutations affected negative regulatory C/EBPε regions I or II (Fig 5F), previously described to restrain C/EBPε activity (Angerer et al, 1999). Taken together, these data show that biallelic *Cebpe* inactivation in dKO cells occurred only in the presence of C/EBPβ LAP*, but not in its absence. In the absence of ectopically expressed C/EBPβ LAP*, either the WT genotype persisted, or alternatively, in-frame deletions affecting negative regulatory C/EBPε regions were selected, strongly supporting the requirement of functional C/EBPε in the absence of *Cebpa/Cebpb*. Accordingly, we conclude that myelo-monocytic *MLL-ENL* transformation remains dependent on C/EBP and that C/EBPε may partially compensate for the loss of *Cebpa/Cebpb*.

## C/EBPs coordinate the expression of the *MLL-ENL/Hoxa* target genes

The combined deletion of *Cebpa* and *Cebpb* led to widespread leukemic cell death and selected for compensatory expression of C/EBPε in dKO clones, suggesting that maintenance of the transformed cell identity is C/EBP-dependent. To test this hypothesis, we performed RNA-seq (two clones, #5, #13, in triplicates) of the *MLL-ENL*–transformed cell transcriptomes before and after compound *Cebpa/Cebpb* deletion. Deleting *Cebpa/Cebpb* resulted in the up-regulation of 2066 (clone #5) and 2,517 (clone #13) genes (755 genes overlapping), respectively, and the down-regulation of 2,447 (clone #5) and 2,702 (clone #13) genes (1,024 genes overlapping),

**Figure 4. Complementation of Growth Defects by C/EBPα and C/EBPβ Isoforms in *MLL-ENL*–Transformed dKO Cells.**
**(A)** Transcripts of *Cebpa* and *Cebpb* with alternative translation initiation sites (arrowheads; cyan) giving rise to different protein isoforms (indicated underneath, purple, and on the right) as examined in Fig 4B. Below: C/EBP expression controls of retrovirally expressed FLAG-tagged C/EBP isoforms expressed in *MLL-ENL*–transformed dKO cells. Cell lysate immunoblots (IB) were probed with anti-FLAG. GFP: negative control retrovirus; C/EBPα isoforms p42, p30; C/EBPβ isoforms LAP*, LAP, and LIP. **(B, C, D)** C/EBPα complementation: (B, C, D). **(E, F, G)** C/EBPβ complementation: (E, F, G). **(B, C, D, E, F, G)** Left, bottom insert:

Color code of complementation assays, as shown in (B, C, D, E, F, G). **(B)** Number of *MLL-ENL*–transformed colonies derived from WT[FL] or dKO cells with and without complementation by retrovirally encoded *Cebpa* p30. Cells were seeded at 500 cells per 35-mm well. The significance of the change in colony number between the dKO and other groups was evaluated with the Kruskal–Wallis test, followed by the post hoc Dunn test. **(C)** Growth curves of WT[FL], dKO, and dKO cells complemented with p30 C/EBPα. **(D)** WST-1 assay of WT[FL], dKO, and dKO cells complemented with p30 C/EBPα. Values are the mean ± SD. The significance of the change in metabolism level between the dKO and other groups was evaluated with the Kruskal–Wallis test, followed by the post hoc Dunn test. **(B, C, D, E)** Colony formation of *MLL-ENL* cells as in (B, C, D). Panel shows the number of *MLL-ENL*–transformed colonies formed in methylcellulose medium by WT[FL], dKO, and complementation by retrovirally encoded C/EBPβ LAP*, LAP, and LIP isoforms. Cells were seeded at 500 cells per 35-mm well. The significance of the change in colony number between the dKO and other groups was evaluated with the Kruskal–Wallis test, followed by the post hoc Dunn test. **(F)** Growth curves of *MLL-ENL*–transformed WT[FL], dKO, and LAP*-, LAP-, LIP-C/EBPβ complementation dKO cells. **(G)** WST-1 assay of *MLL-ENL*–transformed WT[FL], dKO, and LAP*-, LAP-, and LIP-C/EBPβ complementation dKO cells. Values are the mean ± SD. The significance of the change in metabolism level between the dKO and other groups was evaluated with the Kruskal–Wallis test, followed by the post hoc Dunn test.

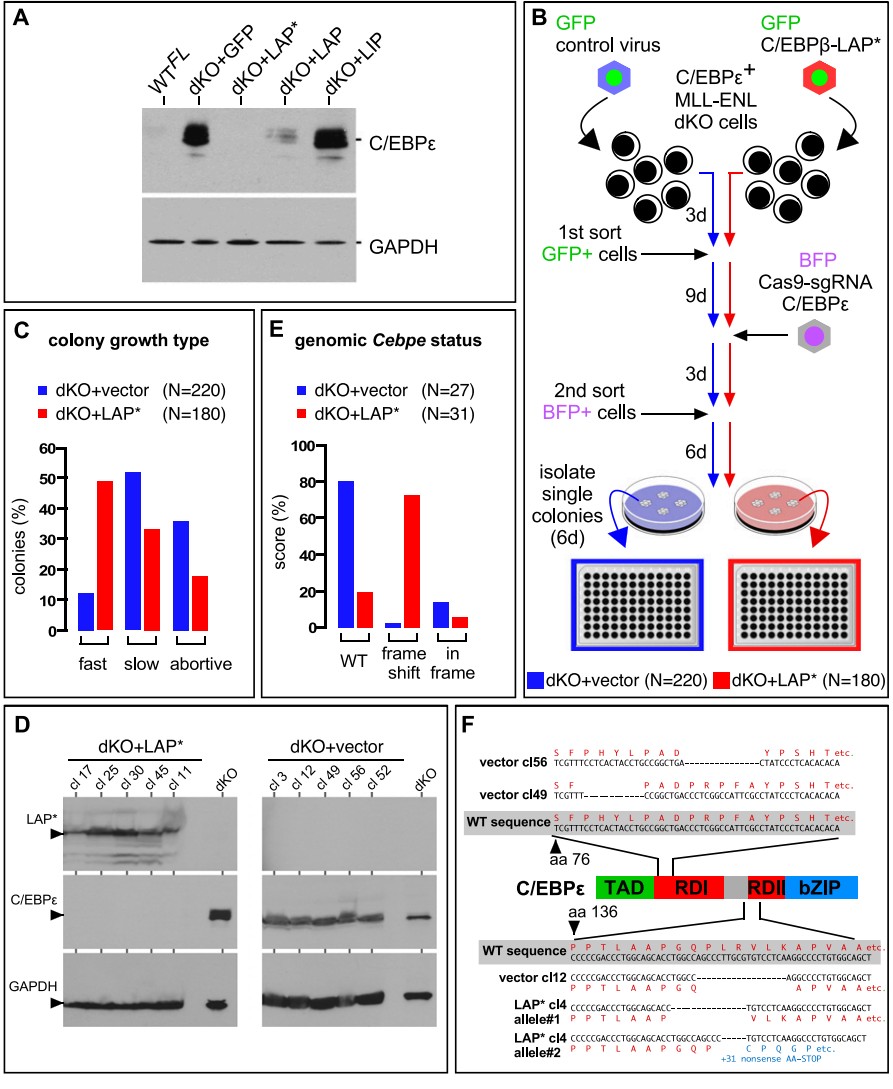

**Figure 5. Examination of C/EBPε dependency of *MLL-ENL*–transformed *Cebpa/Cebpb* dKO C/EBPε⁺ cells via genome editing.**
**(A)** C/EBPε protein levels after prolonged growth of WT^FL and dKO cells infected with control virus (GFP) or vectors encoding different C/EBPβ isoforms. **(B)** Scheme of Cas9-mediated genome editing. C/EBPε⁺ dKO cells were infected with control virus or C/EBPβ LAP*-encoding virus, both expressing GFP. After infection, cells were grown for 3 d, sorted by flow cytometry, and GFP⁺ cells were expanded for 9 d. C/EBPε⁺ dKO vector control cells and C/EBPβ LAP*-complemented cells were infected with retrovirus encoding BFP, Cas9, and guide RNAs targeting *Cebpe* exon 1. After 3 d, the cells were sorted and expanded for 6 d, and seeded in semi-solid medium containing IL-3. Individual colonies were isolated and expanded in 96-well plates in liquid medium (plus IL-3). **(C)** Growth of isolated cells from single colonies was microscopically inspected over 1 wk and scored as fast, slow, or abortive growth. **(D)** Randomly chosen fast-growing clones from (D) were examined by protein blotting for expression of C/EBPε and C/EBPβ, and GAPDH as the loading control by protein-specific antibodies, as indicated on the left. **(E)** *Cebpe* exon 1 DNA derived from 27 vector control clones and 31 C/EBPβ LAP*-complemented clones was amplified by PCR and analyzed by Sanger sequencing and the Inference of CRISPR Edits (ICE) software tool (ice.synthego.com). **(F)** (Middle) Scaled schematic representation of C/EBPε showing the transactivation (TAD), regulatory (RD1 and RD2), and basic zipper (bZIP) regions. Top: examples of in-frame deletions in RD1 from two vector control clones (clones #49 and #56). Bottom: examples of in-frame deletions in RD2 (vector control clone #12) and one LAP*-complemented clone (#4) with an in-frame deletion in one C/EBPε allele and frameshift in the second allele.

respectively, when a corrected *P*-value < 0.05 was used as the cutoff (Fig 6A and Table S1).

We first investigated the impact of the combined loss of C/EBPα and C/EBPβ on the direct and indirect targets of *MLL-ENL* (Collins et al, 2014; Garcia-Cuellar et al, 2016). Of the 166 direct/core *MLL-ENL* target genes, a significant 70 genes ($P = 2.97 \times 10^{-12}$) were among the deregulated genes. Interestingly, 61 of these 70 genes had absolute fold change (FC) < 2, including *Hoxa10* and *Hoxa9* ($P = 0.07$) (Fig 6B). As many of the *MLL-ENL* target genes were significantly but not highly deregulated, we presumed that the high levels of surrogate C/EBPε stabilized the expression of the core *MLL-ENL* target genes in C/EBPε⁺ dKO. Other, more dysregulated *MLL-ENL* candidates (FC > 2; *P* < 0.05) were the DOT1L/transcription initiation–sensitive genes *Bahcc1* and *Fut8*, and the Brd4/elongation-sensitive genes *Sox4* and *Mpo*, which were all up-regulated, suggesting that they were partially repressed by C/EBPβ and/or C/EBPα (as already known for C/EBPα-mediated *Sox4* suppression [Zhang et al, 2013]). Otherwise, C/EBPε may represent a less potent inhibitor or a more potent activator of these genes (Zhang et al, 2013).

For the indirect *MLL-ENL* targets, removing C/EBPα and C/EBPβ affected the genes downstream of the transcriptional control of *HOXA9/MEIS1* that had been identified after the removal of *Cebpa* in *HOXA9/MEIS1*–transformed cells (Collins et al, 2014; Collins & Hess, 2016a, 2016b). Gene set enrichment analysis (GSEA) (Liberzon et al, 2011; Wu & Smyth, 2012) revealed significantly deregulated HOXA9-mediated gene suppression (*P* < 0.05) (Fig 6C), including *Cpa3*, *Gzmb*, *Peg13*, *Mycn*, *Hgfac*, *Alox5*, *Gata2*, *Nkg7*, *Stx3*, *Cdh17*, *Scin*, *Rgs10*, *Havcr2*, and *Col18a1*, but excluding the self-renewal inhibitor *Cdkn2b* (gene set extracted from Collins et al [2014]). The entire HOXA9/C/EBPα co-activated gene set was not significantly deregulated, yet the dKO cells showed increased *Adra2a*, *Pcp4l1*, *Itsn1*, and *Igf2r* expression from the HOXA9/C/EBPα co-activated set, whereas *Adam17*, *Gm1110*, *Pde7a*, *Pdcd4*, and *Nrg2* expression were reduced.

Next, we investigated lineage restriction in the transformed cells by comparing the transcriptional landscape of WT^FL and dKO cells to the ImmGen database (Heng & Painter, 2008). Compared with the WT^FL cells, the dKO cells lost some of their myeloid expression

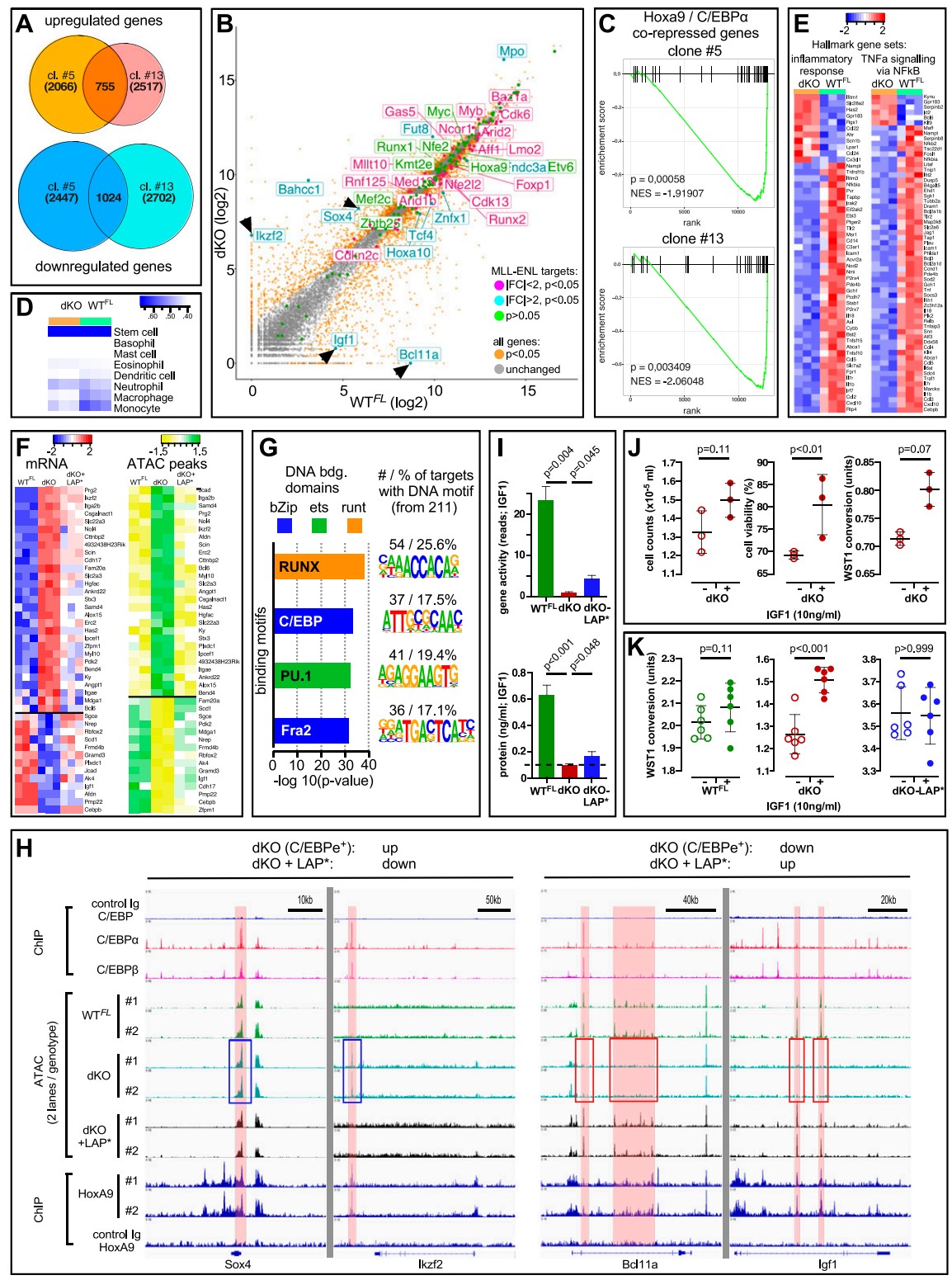

**Figure 6. Molecular Genetic Profiling of *MLL-ENL*–Transformed dKO Cells.**
**(A)** Venn diagrams showing the overlap between up-regulated (Benjamini–Hochberg corrected *P*-value < 0.05) and down-regulated (Benjamini–Hochberg corrected *P*-value < 0.05) genes in both dKO clones as determined by RNA-seq. N = 3 per genotype. **(B)** Scatter plot showing dysregulation of *MLL-ENL* target genes, which are colored according to *P*-value and fold change. All remaining genes are colored according to their *P*-value only (see figure key). Two genes, *Igf1* and *Bcl11a*, are included in the plot, as they are known *MLL* translocation targets. **(C)** Gene set enrichment analysis showing significant enrichment of genes co-repressed by HOXA9/C/EBPα among the deregulated genes. Most of these genes were up-regulated in the dKO cells. **(D)** Comparison of expression patterns to that of ImmGen data shows a shift in overall gene

characteristics and showed less similarity to monocytes and macrophages (Fig 6D). Specifically, *Csf1r*, Rel family members (*Rel*, *Rela*, and *Relb*), and *Tgfbr2* levels were decreased in the dKO cells, supporting the pioneering role of C/EBPα and C/EBPβ in myeloid commitment (Wang et al, 2006; Jaitin et al, 2016; Tamura et al, 2017). However, the WT^FL and dKO cells were similarly enriched for genes shared with HSC, highlighting the LSC state of the cells.

For a more general/unbiased approach, we performed GSEA on the hallmark gene sets of MsigDB to examine the enrichment of functional terms in the group of differentially expressed genes between the WT^FL and dKO genotypes (Liberzon et al, 2011; Wu & Smyth, 2012). Consistent with the reduced myeloid commitment observed in the dKO cells (Fig 2A), we detected strong dysregulation of the inflammatory genes, documented by significant enrichment of "hallmark inflammatory response" and "hallmark Tnfa signaling via Nfkb" (*P* < 0.05) with partially overlapping genes, in addition to an increase in stem cell transcripts (see the corresponding heat maps in Fig 6E). Altogether, these results support the premise that C/EBPε partially compensates for the loss of *Cebpa/Cebpb* by securing a MLL-ENL and Hoxa9 regulated core gene expression program and shifts the transformed phenotype towards an immature phenotype.

The superior growth and viability of the WT^FL and dKO-LAP* cells over *Cebpa/Cebpb* dKO cells was evident, affirming that compensation by C/EBPε remains incomplete. To identify ancillary *MLL-ENL* targets that may be involved in restoring the WT phenotype, we compared differential gene expression with the genome accessibility status using Assay for Transposase-Accessible Chromatin with sequencing (ATAC-seq) (Buenrostro et al, 2013) in the three genotypes, that is, WT^FL, *Cebpa/Cebpb* dKO, and *Cebpa/Cebpb* dKO complemented with C/EBPβ LAP*. Initial inspection of chromatin accessibility around the *Cebpa/Cebpb* loci confirmed the dKO status, whereas the ATAC peak pattern around the *Cebpe* locus remained unchanged, ruling out the possibility that a recombination event at the *Cebpe* locus may have contributed to enhanced *Cebpe* expression in the dKO cells (Fig 6H). Focusing on genes that exhibited both, re-establishing gene expression and restoring chromatin status in dKO-LAP* cells towards WT^FL status yielded a list of genes (highly correlated gene expression and chromatin status), part of which is shown in Fig 6F. Importantly, known motif search showed significant enrichment of the CEBP, ETS, and RUNX binding motifs (Fig 6G) in the genomic regions determined by the full list of ATAC peaks, as described in Fig 6F. Fig 6H shows that, in dKO cells, the enhanced *Sox4* and *Ikzf2* gene expression (Fig 6B) was associated with increased chromatin accessibility, whereas decreased *Bcl11a* and *Igf1* gene expression was associated with

decreased ATAC peak sizes, whereas ectopic expression of LAP* restored ATAC peaks to the WT conditions (Fig 6H).

Open chromatin regions that were present in WT^FL cells, absent in dKO cells, and restored in dKO-LAP* cells were identified in an intronic region of the *Igf1* gene (Fig 6H) that overlapped with a super-enhancer region previously identified in monocytes and liver cells (Jiang et al, 2019). Moreover, IGF1 has been described as a growth-regulatory *MLL-ENL/Hoxa9/Meis1* target and as a direct C/EBPβ target gene involved in autocrine stimulation of transformed murine monocytes and in neoplasia (Pollak et al, 2004; Wessells et al, 2004; Steger et al, 2015; Collins & Hess, 2016b). Comparison of the *Igf1* RNA-seq reads confirmed *Igf1* gene expression in the WT^FL and dKO-LAP* cells, but not in the dKO cells (Fig 6I, top panel), and quantification by ELISA in tissue culture supernatants (Fig 6I, bottom panel) showed that IGF1 was produced by the WT^FL and dKO-LAP* cells, but not by dKO cells. Adding IGF1 (10 ng/ml) to the culture medium increased the cell number, viability (toluidine exclusion), and metabolism (WST-1 conversion) of the dKO culture (Fig 6J) but not that of the WT^FL or dKO-LAP* cultures (Fig 6K). Taken together, these data suggest that C/EBP family member- and isoform-specific regulation can differentially affect several *MLL-ENL* target genes. The data also confirm a C/EBPβ co-regulated autocrine function of IGF1 in *MLL-ENL* transformation.

## Discussion

Myelomonocytic commitment and differentiation into the GMP state depends on C/EBP transcription factors and is a prerequisite for the emergence of AML and LSC. Our data show that the continuous presence of C/EBP transcription factors is essential for maintaining leukemic myelomonocytic transformation by the *MLL–ENL* oncogene product of fetal liver hematopoietic progenitor cells in tissue culture.

A current concept of myeloid leukemogenesis suggests that LSC arise in committed progenitors that resemble GMP-like cells that acquire self-renewal capacity (Cozzio et al, 2003; Huntly et al, 2004; Jamieson et al, 2004; Krivtsov et al, 2006). As an inducer of the GMP-like state, it has been suggested that C/EBPα plays a critical role in leukemic myelomonocytic transformation. Intriguingly, once leukemic progenitor transformation is established, C/EBPα can be genetically removed, whereas the leukemic state persists (Ohlsson et al, 2014; Ye et al, 2015). Along the same lines, the initiation of oncogene-induced leukemic transformation in the absence of C/EBPα

expression patterns, with loss of myeloid characteristics and maintenance of stem cell patterns in dKO cells indicated by Spearman correlation. **(E)** Heatmap of inflammatory and TNFα/NF-κB hallmark genes (gene sets derived from MSigDB) in WT^FL and dKO cells. Both terms are significantly enriched (adjusted *P*-value < 0.05, according to Gene set enrichment analysis). The heat maps show genes of the respective gene sets exhibiting an adjusted *P*-value < 0.05 and an absolute fold change > 2. Row $z$-score of normalized $\log_2$ counts. **(F)** Subset of the co-regulated transcriptome and genome structure as determined by RNA-seq and ATAC-seq. The whole list comprises genes that exhibit both the re-establishment of gene expression and restoration of the chromatin status in dKO-LAP* cells to WT^FL status. Row $z$-score of normalized $\log_2$ counts. **(F, G)** Known motif prevalence in dKO-LAP* cells in the full set of genes shown in (F) as determined by HOMER. **(H)** Examples of replicate ATAC-seq data from WT^FL, dKO, and dKO-LAP* *MLL-ENL*–transformed cells indicating gain of peaks (left, blue outline) in dKO at the *Sox4* and *Ikzf2* loci, and loss of peaks at the *Bcl11a* and *Igf1* loci (right, red outline). The top three lanes show C/EBPα and C/EBPβ chromatin immunoprecipitation (ChIP) data from *MLL-AF9*–transformed cells (Roe et al, 2016); the bottom three lanes show *Hoxa9* ChIP data from Zhong et al (2018). **(I)** *Igf1* gene expression (top: RNA-seq data from triplicates) and IGF1 secretion into the growth medium (bottom: ELISA data; ELISA was repeated 3 times yielding similar results) by *MLL-ENL*–transformed cells. **(J)** Response of *MLL-ENL*–transformed dKO cells to the addition of IGF1 to the culture medium as determined by cell counts, viability, and metabolic activity. **(K)** Comparison of response to IGF1 of WT^FL, dKO, and dKO-LAP*-complemented cells.

can still be achieved, if concomitant emergency granulopoiesis is induced (Ye et al, 2015). These data suggest the achievement of the GMP-like state as the most critical event in leukemic transformation, regardless of how the GMP state is achieved. However, emergency granulopoiesis elicits C/EBPβ expression (Hirai et al, 2006), and gene replacement experiments have shown that C/EBPβ expressed from the C/EBPα locus can fully complement the hematopoietic functions of C/EBPα, including differentiation into GMP (Jones et al, 2002). The alternative explanation for the maintenance of the leukemic state after removing C/EBPα, therefore, is that C/EBPα may initially pass the "GMP baton" to C/EBPβ, which in turn supports proliferation and maintains the transformed myeloid state.

Our data show that maintaining myeloid *MLL-ENL* transformation strongly depends on combined C/EBPβ and C/EBPα expression. The non-oncogenic dependence (Solimini et al, 2007; Nagel et al, 2016) on C/EBP transcription factors became evident from the analysis of the dKO mutants, which yielded very few clonogenic cells that consistently expressed resident C/EBPε. Genetic removal of both *Cebpa* and *Cebpb* consequently revealed a surrogate function of C/EBPε in myeloid *MLL-ENL* transformation. This finding is in line with several KO studies involving various members of the C/EBPα, β, δ, and ε family that revealed redundant C/EBP functions only in compound deletion mutants. Indeed, C/EBPε partially compensated for the loss of C/EBPβ during murine myelopoiesis and the innate immune response and was also compatible with the proliferative progenitor state (Akagi et al, 2010; Cirovic et al, 2017). Whether the rare event of C/EBPε expression in the *MLL-ENL*–transformed cultures may have occurred before or after *Cebpa* or *Cebpb* deletion, potentially reflecting *MLL-ENL* progenitor plasticity, or as a regulatory event that occurred under the selective pressure of *Cebpa*/*Cebpb* removal, remains to be determined.

The data suggest a compensatory backup function of C/EBPε in the surviving *Cebpa*/*Cebpb* dKO *MLL-ENL* cells that may maintain the minimal essential GMP condition in the absence of *Cebpa*/*Cebpb* and serve as a proxy in regulating the most critical genes involved in *MLL-ENL* transformation. Consistently, genetic removal of C/EBPε from dKO cells by gene editing was only possible after complementation with C/EBPβ LAP*, whereas uncomplemented cells only tolerated C/EBPε gene editing as in-frame mutations that removed parts of the negative regulatory regions of the C/EBPε protein (Angerer et al, 1999). Taken together, the results suggest the essential functions of C/EBP family members, and in particular, a privileged function of C/EBPβ LAP* in the proliferation, survival, and maintenance of the myeloid cell fate of transformed *MLL-ENL* cells.

Regulation of the Hox cluster target genes, including *Hoxa9* and *Meis1*, previously emerged as a hallmark of *MLL* translocation leukemia, and the combined expression of *Hoxa9* and *Meis1* was sufficient for initiating a leukemogenic progenitor self-renewal program (Armstrong et al, 2002; Ayton & Cleary, 2003; Ferrando et al, 2003; Huang et al, 2012; De Braekeleer et al, 2014; Garcia-Cuellar et al, 2016; Collins & Hess, 2016a). C/EBPα is also essential in *Hoxa9*/*Meis1*-induced leukemic transformation (Collins et al, 2014), yet *HOXA9*/*HOXA10*/*Meis1* were only partially down-regulated in the *Cebpa*/*Cebpb* dKO C/EBPε+ cells. We believe that the expression of resident C/EBPε in the dKO cells compensated further down-regulation beyond a lower limit, thereby functionally substituting for the absence of *Cebpa*/*Cebpb*, whereas cells that failed to express C/EBPε were lost.

Accordingly, the most strongly affected genes in the dKO cells may therefore represent secondary functional targets, such as autocrine IGF1 stimulation, which contributed to accelerated growth and prevention of apoptosis, but may not be entirely central to leukemic transformation.

Previous research suggesting a C/EBPα–induced epigenetic state that is subsequently maintained by the MLL-ENL oncoprotein (Ohlsson et al, 2014; Roe & Vakoc, 2014) is contrasted by various lines of evidence suggesting that, in higher organisms, maintenance of the epigenetic state relies on the continuous presence of priming master regulatory transcription factors (Blau et al, 1985; Blau & Baltimore, 1991; Holmberg & Perlmann, 2012). Our data support the latter concept and suggest a prominent role for distinct C/EBP family members and isoforms in *MLL-ENL* transformation, reflecting a type of lineage- and differentiation state–specific non-oncogenic dependence on transformation. Pharmacological interference with the non-oncogenic C/EBP dependence on acutely transforming myelomonocytic oncoproteins could potentially extend a future clinical repertoire of treatment regimens with the aim of eradicating devastating infant AML.

# Materials and Methods

## Generation of *MLL-ENL*–transformed cells

Viral supernatants were produced by transfecting Plat-E cells with pMIG-GFP retroviral vector or pMSCV-MLL-ENL construct. Infectious supernatant was collected 48 and 72 h post-transfection. Mouse embryos (13.5 d) were dissected and genotyped. Hematopoietic stem and progenitor cells were isolated from the mouse fetal livers and transduced during two consecutive days by centrifugation (twice: 1 h, 900*g*) with viral supernatant and 8 μg/ml hexadimethrine bromide. After the second infection, the cells were recovered in IMDM supplemented with murine IL-3 (10 ng/ml), IL-6 (10 ng/ml), and SCF (50 ng/ml) (STEMCELL Technologies). Infected cells were selected with 500 μg/ml G418 (Sigma-Aldrich). After 14-d antibiotic selection, the cells were seeded in MethoCult M3434 (STEMCELL Technologies). Colonies formed from single cells were isolated, expanded, and analyzed. Flow cytometry was performed with LSR II or Fortessa machines (Becton Dickinson). The following antibodies from BioLegend or eBioscience were used: CD11b (M1/70), CD117 (2B8), Ly6C (AL-21), Ly6G (1A8), CD115 (AFS98), Sca1 (D7), and CD64 (X54-5/7.1).

## Cell culture

The *MLL-ENL*–transformed cell lines were cultured in IMDM (10% FCS, 1% penicillin/streptomycin [Gibco]) supplemented with 10 ng/ml mouse IL-3 (STEMCELL Technologies). Cells were split every 2–3 d to $1 \times 10^5$ cells/ml. Plat-E cells were cultured in DMEM (10% FCS, 1% Hepes, and 1% penicillin/streptomycin [Gibco]). Transient transfections were performed using polyethylenimine (Polysciences) according to the manufacturer's protocol.

## TAT-Cre treatment and selection of recombinant clones

Before the treatment, TAT-Cre recombinase (Millipore) was diluted to a final concentration of 4 μM in IMDM and filtered (0.2-μm

low-protein-binding filter). Cell pellets of $4 \times 10^5$ *MLL-ENL*–transformed cells were resuspended in 1 ml TAT-Cre solution, transferred to a 24-well plate, and incubated at 37°C for 20 h. The cells were washed with PBS and seeded in MethoCult M3134 (STEMCELL Technologies) supplemented with 10 ng/ml mouse IL-3. Colonies were transferred to liquid medium, expanded, and analyzed by PCR for excision of *Cebpa* and/or *Cebpb*.

## Retroviral transduction with C/EBP isoforms, GFP vector control, or BFP Cas9-sgRNA C/EBPε

Viral supernatants were produced by transfecting Plat-E cells with empty pMIG-GFP retroviral vector or pMIG-GFP constructs containing p42, p30, LAP*, LAP, or LIP, or pMSCV-BFP constructs containing Cas9 and sgRNAs targeting *Cebpe*. The cells were centrifuged with infectious supernatant collected 72 h after transfection and 8 μg/ml hexadimethrine bromide (1 h, 900*g*), and left for recovery overnight.

## Plasmids and retroviral constructs

The *MLL-ENL* retroviral construct has been described previously (Slany et al, 1998). C/EBPα, β isoform expression constructs, amino acid sequence, and retroviral construction have been published previously and are available on request (Kowenz-Leutz et al, 1994; Stoilova et al, 2013; Cirovic et al, 2017). pMSCV_Cas9-2A-GFP-sgRNA (#124889; Addgene) and pMSCV-U6sgRNA(BbsI)-PGKpuro2A-BFP (#102796; Addgene) were purchased (Li et al, 2016; Henriksson et al, 2019). The T2A-GFP fragment of the pMSCV GFP-Cas9–containing construct was excised by *Bam*HI/*Not*I restriction digest. The T2A-BFP marker of pMSCV-U6sgRNA(BbsI)-PGKpuro2A-BFP was amplified by PCR and cloned as a *Bam*HI/*Not*I fragment into the pMSCV Cas9 construct. BFP primer: 5′ BamHI_T2A-BFP 5′-CGCGGATCCGGAGAGGGCAGAGGAAGTCTC-3′; 5′-BFP-NotI: 5′-ATAGTTTAGCGGCCGCTCAATTAAGCTTGTGCCC-3′.

Cebpe sgRNAs were selected using CrispRGold (https://crisprgold.mdc-berlin.de) and CRISPOR (http://crispor.tefor.net). The sgRNAs were cloned by BbsI into the pMSCV_Cas9-2A-BFP-sgRNA vector. The C/EBPε sgRNA oligos used were 5′-CTTACCTTGAGGACACGCAA-3′, 5′-AGGGATAGGCGAATGGCCGA-3′, 5′-CTCGTTTCCTCACTACCTGC-3′, and 5′-CGCACTCATAGTAGGTCCCG-3′.

## Analysis of C/EBPε genome editing by CRISPR/Cas9 and C/EBPε sgRNAs

Total DNA was isolated from single cell clones using Quick-Extract DNA Extraction Solution (Epicentre). *Cebpe* genomic exon 1 was amplified by PCR, the fragments were purified (Invisorb #1020300300; Invitek), and analyzed by Sanger sequencing (LGC Genomics). The edited sequences were analyzed using modified ICE software (Hsiau et al, 2018 Preprint), available from Synthego (https://www.synthego.com/products/bioinformatics/crispr-analysis). The PCR primers used for *Cebpe* exon 1 analysis were 5′-CAGGACA-CAGCCGAGTTCTA-3′ and 5′-CTAGGGCAAATCTAGGACCG-3′.

## Genotyping

Total DNA was isolated using QuickExtract DNA Extraction Solution (Epicentre). *Cebpa* excision was evaluated by two separate PCR for the *Cebpa* flox allele and *Cebpa* deletion. The *Cebpa* flox allele primers

were 5′-TGGCCTGGAGACGCAATGA-3′ and 5′-CGCAGAGATTGTGCGTCTTT-3′; the expected product was 269 bp. The *Cebpa* deletion primers were: 5′-GCCTGGTAAGCCTAGCAATCCT-3′ and 5′-TGGAAACTTGGGTTGGGTGT-3′; the expected product was 380 bp. *Cebpb* excision was determined by competitive PCR using the following primers: 5′-GAGCCACCGCGTCCT-CCAGC-3′, 5′-GGTCGGTGCGCGTCATTGCC-3′, and 5′-AGCAGAGCTGCCCC-GGCAAA-3′; the reaction produced bands of 253 bp for the flox allele and 610 bp for the deleted allele.

## Immunoblotting

Total protein lysates were prepared by lysing fresh cell pellets with 0.5 M NaOH with subsequent neutralization with 0.5 M HCl. The samples were sonicated, mixed with SDS sample buffer and glycerol, and heated (3 min at 95°C). After centrifugation, the protein lysates were separated by electrophoresis and transferred to a nitrocellulose membrane using a Trans-Blot Turbo System (Bio-Rad). Protein signals were detected after incubation with antibodies against C/EBPα (14AA; Santa Cruz Biotechnology), C/EBPβ (C-19; Santa Cruz Biotechnology), C/EBPδ (C-22; Santa Cruz Biotechnology), C/EBPε (NBP1-85446; Novus Biologicals), Flag-M2 (F-1804; Sigma-Aldrich), and GAPDH (ab9484; Abcam).

## Apoptosis assay

The level of apoptosis was determined with a FITC–Annexin V Apoptosis Detection Kit I (BD Pharmingen). After washing twice with cold PBS, $1 \times 10^6$ cells were resuspended in 1 ml 1× binding buffer. Cell suspension (100 μl) was transferred to a new tube and mixed with 5 μl FITC–annexin V. The samples were incubated for 15 min at room temperature in the dark. Afterward, 5 μl 7-AAD (BD Pharmingen) was added and the samples were subsequently analyzed using an LSR II flow cytometer (Becton Dickinson).

## Proliferation and metabolism assays

The *MLL-ENL*–transformed cells were seeded at $5 \times 10^4$ cells per well in 100 μl complete IMDM in 96-well plates in triplicate. The cells were incubated for 48, 72, or 96 h at 37°C. The cell proliferation reagent WST-1 (10 μl/well; Roche) was added, gently mixed, and incubated at 37°C for 1 h. The absorbance of the samples was measured at 450 nm using an iMark microplate reader (Bio-Rad). A murine IGF-1 ELISA kit (BGK9PU89; BioGems) was applied according to the manufacturer's protocol. For the CFSE proliferation assay, $1 \times 10^6$ cells were resuspended in 1 ml 5 μM CFSE solution (BioLegend) for 20 min in the dark at room temperature. Afterward, the reaction was quenched using 5 ml cell culture medium plus 10% FCS. The cells were washed and cultured for the indicated time.

## Colony formation assay, determination of colony size and number, serial replating

The *MLL-ENL*–transformed cells were added to complete (10% FCS, 1% penicillin/streptomycin [Gibco]) and IL-3–supplemented (10 ng/ml) MethoCult 3134 (STEMCELL Technologies) at a final concentration of 5,000 cells per mL (if not indicated otherwise). The cell mix (1 ml) was seeded in six-well meniscus-free plates (SmartDish;

STEMCELL Technologies) in triplicate. High-resolution microscopic scans were taken after 7, 10, and 14 d using an EVOS imaging system (Thermo Fisher Scientific). The number, size, and approximate cell numbers of the colonies were calculated using ImageJ software with the ColonyArea plugin. Serial replating experiments were set up with 5,000 cells per mL in MethoCult 3134 supplemented with IL-3 (10 ng/ml), IL-6 (5 ng/ml), SCF (25 ng/ml) and GM-CSF (5 ng/ml). After 7 d in culture, colony numbers were determined using the EVOS imaging system, colonies were harvested by centrifugation in culture medium, cell numbers and viability were determined, and comparable numbers of cells were re-seeded in MethoCult 3134 under the conditions described above.

### ATAC-seq

To prepare the ATAC-seq libraries, 50,000 cells were sorted into 500 $\mu$l PBS. The libraries were prepared as previously described (Buenrostro et al, 2013), with slight modifications (Lara-Astiaso et al, 2014). Briefly, the cells were lysed with 25 $\mu$l cold lysis buffer (10 mM Tris–HCl [pH 7.4], 10 mM MgCl$_2$, and 0.1% Igepal CA-630) and the nuclei were pelleted by centrifugation for 25 min at 4°C and 500$g$ in a swing rotor with low acceleration and brake settings. The pellet was resuspended in 25 $\mu$l reaction mix containing 2 $\mu$l Tn5 transposase and 12.5 $\mu$l TD buffer (Nextera DNA library preparation kit; Illumina), and incubated at 37°C for 1 h. Next, 5 $\mu$l cleanup buffer (900 mM NaCl, 30 mM EDTA) with 2 $\mu$l 5% SDS and 2 $\mu$l proteinase K (NEB) were added, and the samples were incubated at 40°C for 30 min. Subsequently, the DNA was purified using AMPure XP beads (Beckman Coulter) and PCR-amplified with KAPA HiFi HotStart ReadyMix (Kapa Biosystems) and indexing primers published previously (Buenrostro et al, 2013). Then, the DNA library fragments were selected for <600-bp fragments and purified using AMPure XP beads. The concentration and fragment size of the final libraries were measured using a Qubit fluorometer (Life Technologies) and TapeStation (Agilent Technologies). The samples were sequenced with an average of 25 million reads per sample using a NextSeq 500 system (Illumina).

### Sequencing data analysis

RNA-seq libraries were prepared using Illumina TruSeq stranded mRNA kit starting with 500 ng of input total RNA. Libraries included dual indices and were equimolar pooled. Paired-end 76 nt sequencing was performed on Illumina NextSeq 500 High Output v.2 flowcell. RNA-seq samples were aligned to the reference genome (mm10) using STAR (Dobin et al, 2013). Expression was quantified using HTSeq (Anders et al, 2015); downstream analysis was done with DESeq2 using the included normalization strategy (for details see Love et al [2014]) and a Benjamini–Hochberg–corrected $P$-value of 0.05 as a significance cutoff. Alterations in function induced by the different expression patterns in the conditions were assessed using GSEA using gene sets by MSigDB (Liberzon et al, 2011) and Camera (Wu & Smyth, 2012). ATAC-seq samples were preprocessed with PiGx pipelines (Wurmus et al, 2018), including alignment to a reference with Bowtie 2 (Langmead & Salzberg, 2012), to genome version mm10 and peak-calling using MACS (Zhang et al, 2008). Differential peaks were detected with DiffBind (2.14) using a corrected $P$-value of 0.05 as a cutoff (Ross-Ines et al, 2012). Known motif detection was performed using findMotifsGenome.pl in HOMER (v4.10.3) ($ findMotifsGenome.pl XXX.bed mm10 homer/ -size

200 -mask) (Heinz et al, 2010). The set of genomic regions used for motif discovery comprises the regions of those genes, that exhibited both, re-established gene expression (uncorrected $P$-value < 0.05) and re-stored chromatin status (corrected $P$-value < 0.05) in dKO-LAP* cells toward the WT$^{FL}$ status.

## Data Availability

Data generated during this study have been deposited in Gene Expression Omnibus with the accession codes GSE153622, GSE153623, and GSE153624.

## Supplementary Information

## Acknowledgements

Reagents were kindly provided by Klaus Rajewsky. We thank Hans-Peter Rahn and Linh Thuy Nguyen for help with flow cytometry; Martin Janz, Stephan Mathas (all Max-Delbrueck-Center for Molecular Medicine, MDC, Berlin), and Karl-Heinz Seeger (Charité, Berlin) for discussion; and Ralph Kühn (MDC) for help with CRISPR data analysis. D Dörr was funded by the International MDC-PhD program and K Zimmermann by a grant from the DFG, CRC167, and B11. A Mildner was funded by a Heisenberg Fellowship from the Deutsche Forschungsgemeinschaft (DFG and MI1328).

### Author Contributions

R Wesolowski: data curation and writing—original draft, review, and editing.
E Kowenz-Leutz: resources, data curation, formal analysis, validation, investigation, methodology, and writing—review and editing.
K Zimmermann: data curation, software, formal analysis, and writing—review and editing.
D Dörr: investigation, methodology, and writing—review and editing.
M Hofstätter: validation, investigation, and methodology.
RK Slany: resources, methodology, and writing—review and editing.
A Mildner: data curation, formal analysis, investigation, and writing—review and editing.
A Leutz: conceptualization, resources, data curation, formal analysis, supervision, funding acquisition, validation, investigation, visualization, methodology, project administration, and writing—original draft, review, and editing.

### Conflict of Interest Statement

The authors declare that they have no conflict of interest.

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
