## [Reviewer comments · Life Science Alliance]

Life Science Alliance

Myeloid Transformation by MLL-ENL Depends Strictly on C/EBP

Radoslaw Wesolowski, Elisabeth Kowenz-Leutz, Karin Zimmermann, Dorothea Dörr, Maria Hofstätter, Robert Slany, Alexander Mildner, and Achim Leutz

DOI: <https://doi.org/10.26508/lsa.202000709>

Corresponding author(s): Achim Leutz, Max Delbrück Center for Molecular Medicine and Alexander Mildner, Max-Delbrück-Center for Molecular Medicine

Review Timeline:	Submission Date:	2020-03-19
	Editorial Decision:	2020-05-07
	Revision Received:	2020-09-22
	Editorial Decision:	2020-10-19
	Revision Received:	2020-10-22
	Accepted:	2020-10-23

Scientific Editor: Shachi Bhatt

Transaction Report:

May 7, 2020

Re: Life Science Alliance manuscript #LSA-2020-00709-T

Prof. Achim Leutz
Max Delbrück Center for Molecular Medicine
Tumorigenesis and Cell Differentiation
Robert-Rössle-Strasse 10
Berlin, Berlin 13125
Germany

Dear Dr. Leutz,

Thank you for submitting your manuscript entitled "Myeloid Transformation by MLL-ENL Depends Strictly on C/EBP" to Life Science Alliance. The manuscript was assessed by expert reviewers, whose comments are appended to this letter.

As you will see, the reviewers appreciate your analyses, but also raise some concerns. Given the input received, we would like to invite you to submit a revised version of your work, addressing the concerns of the reviewers. Reviewer #1 points out that a knockout approach to analyze effects of LOF of *cebpe* would significantly strengthen your work, and reviewer #3 also thinks that the data pertaining to *cebpe* are rather weak at this stage. Both reviewer #2 and #3 think that the drug response analysis does not add to the paper and should get removed, please do so. Furthermore, the properties of C/EBPb KO MLL-ENL cells should be further characterized, and the suggestions on changing the data representation followed.

In our view these revisions should typically be achievable in around 3 months. However, we are aware that many laboratories cannot function fully during the current COVID-19/SARS-CoV-2 pandemic and therefore encourage you to take the time necessary to revise the manuscript to the extent requested above. We will extend our 'scooping protection policy' to the full revision period required. If you do see another paper with related content published elsewhere, nonetheless contact me immediately so that we can discuss the best way to proceed.

Please note that papers are generally considered through only one revision cycle, so strong support from the referees on the revised version is needed for acceptance.

Thank you for this interesting contribution to Life Science Alliance. We are looking forward to receiving your revised manuscript.

Sincerely,

B. MANUSCRIPT ORGANIZATION AND FORMATTING:

*****IMPORTANT:** It is Life Science Alliance policy that if requested, original data images must be made available. Failure to provide original images upon request will result in unavoidable delays in publication. Please ensure that you have access to all original microscopy and blot data images

before submitting your revision.***

Reviewer #1 (Comments to the Authors (Required)):

The manuscript by Wesolowski et. al. studies the potential role for cebpb and cebpe in the maintenance of MLL-ENL driven leukemia, particularly in the absence of cebpa. This line of studies is prompted by previous observations that cebpa is required for initiation, but not maintenance of leukemia. They establish a convincing role for cebpb as relevant in this setting and a suggestive role for cebpe. This set of studies provides important understanding of MLL-ENL biology and the role of cebp proteins in the setting of myeloid progenitor transformation. However, there are specific aspects that could be clarified to enhance impact.

Comments:

1. The authors mention the inability to suppress cebpe expression via shRNA or siRNA. This problem is understandable in this system. However, many have been able to inactivate genes by CRISPR mediated approaches in these cells. This would be a very helpful experiment in terms of validating the role of cebpe if it could be completed.
2. It is hard to tell in the manuscript if they ever assessed leukemia generation as opposed to in vitro cell proliferation/survival of the various genotypes. If not, this does influence the interpretation of some of the data and thus needs to be clearly pointed out in the discussion. Some of the results regarding dependency might be different if growth were assessed in vivo. This caveat should be clearly stated.
3. The cells transformed here were from fetal liver whereas most published experiments assess HSC/progenitors from postnatal mice. This could influence results since these cell types are quite different. This should be similarly pointed out in the discussion.
4. The authors seems to propose that cebp proteins and Hox proteins are working together to maintain a leukemia gene expression program. This is reasonable and in line with previous publications. Figure 5C is particularly important. Where did this gene set come from? There should be increased explanation of this figure as much of the argument around the function of cebp relies on this data. This is particularly true given the modest change in Hox gene expression which is nicely pointed out by the authors.

Reviewer #2 (Comments to the Authors (Required)):

The paper of Radoslaw Wesolowski,et al. reported that removing C/EBPb alone or both C/EBPa and C/EBPb slowed the growth and reduced the viability of MLL-ENL-transformed cells; C/EBPe showed increased expression in dKO-MLL-ENL colonies; furthermore ectopic expression of p30 and C/EBPb in C/EBPa/b KO cells partially rescued MLL-ENL-transformed cell growth. Thus the authors concluded that myelomonocytic MLL transformation and its maintenance remains continuously dependent on C/EBP family members.

Suggestions for improvement: major

- 1) It has been reported by multiple groups that C/EBPa is not required by the maintenance of MLL-

ENL transformed cells. In this paper the authors tested the hypothesis that other C/EBP family members can contribute to the maintenance of the leukemic phenotype. In figure 1, the authors examined the role of C/EBPb in leukemic phenotype maintenance and showed loss of C/EBPb did not affect the number of colonies but reduced cell growth and viability. To better understand the role of C/EBPb in MLL-ENL transformation maintenance and to compare C/EBPa:C/EBPb Dko-MLL-ENL cells, it is suggested that the properties of these C/EBPb KO MLL-ENL cells could be further characterized, including colony formation in a serial replating assay (which correlates with in vivo leukemogenesis), cell cycle, apoptosis, and the expression of other C/EBP family members.

2) In Figure 2a the authors compared the response of WT-MLL-ENL cells vs C/EBPa:C/EBPb dKO-MLL-ENL cells to the chemotherapy drug cytarabine, daunorubicin, or both. The authors concluded that dKO-MLL-ENL cells showed more apoptosis and more sensitivity to chemotherapy. However, it is not clear the relevance of this data to the main hypothesis of the paper. Furthermore, although the authors reported dKO-MLL-ENL were more sensitive to chemotherapy, if one looks at the percentage of annexin V+ following cytarabine treatment, the percentage of apoptotic/dead cells in WT MLL-ENL transformed cells increased from 38.7% to 63.5% (1.6 fold) in clone 1 and 16.37% to 64.7% (4 fold) in clone 2. In dKO MLL-ENL cells, the percentage of annexin V cells increased from 48.4% to 82.4% (1.7 fold) in clone 1 and from 45.5% to 85% (1.9 fold) in clone 2 (Figure 2a). Similar results were observed with other treatments. Based on these results, it is hard to conclude that dKO MLL-ENL cells are more sensitive. Mechanistically, cytarabine and daunorubine are targeting proliferating cells, while loss of C/EBPb slowed the proliferation, so it is hard to explain why dKO MLL-ENL should be more sensitive to these chemotherapeutic drugs. Furthermore, it is hard to assess responsiveness to chemotherapy without a good non-leukemic control, it would have been useful to have utilized nontransformed GMP as a control. Again, it is not clear how much this adds to the manuscript.

3) Although the authors reported that removing both C/EBPa and C/EBPb reduced the number of viable MLL-ENL transformed cells (line 118-135), a thorough characterization of C/EBPa:C/EBPb dKO-MLL-ENL cells are necessary. A figure with side by side comparison between C/EBPb KO and dKO leukemic cells would be helpful. These analysis should include serial replating, cell cycle, and apoptosis.

4) The data in Figure 5B data suggested very mild changes in expression of many MLL-ENL target genes following C/EBPa and C/EBPb deletion. It is also interesting to notice that C/EBPa and C/EBPb dKO did not affect the HSC enriched gene signature, an indicator for stemness. Given these results, one would argue that MLL-ENL alone might be sufficient to maintain LSC status, while C/EBPa, C/EBPb or C/EBPe more likely contributes to leukemic cell proliferation, survival and myeloid identity. Could authors comment on this?

5) Figure 5F showed a nice correlation between mRNA expression and chromatin conformation for a few of genes following C/EBP dKO and dKO-LAP rescue. How does this correlation apply to other up and down genes?

Minor Suggestions

1) Line 133: "14 Cebpa/Cebpb dKO subclones from two mice (2 and 14 subclones, respectively) ". Is (2 and 14) a typo? Should it be (2 and 12)?

2) Figure 4E, the color label did not match with the color labels in Figure 4A

Reviewer #3 (Comments to the Authors (Required)):

Summary:

Previous work demonstrated that C/EBP α is essential for MLL-ENL induced transformation, however, once transformation has taken place, its expression is not required for leukaemia maintenance. Wesolowski et colleagues explored the possibility that other members of the C/EBP transcription factor family are required to sustain the tumour once it is established. The authors show that Cebpb or combined Cebpa/Cebpb gene deletion affects growth and viability of established leukemic cells in an MLL-ENL-transformed murine cell model. Simultaneous Cebpa/Cebpb knock-out (dKO) augments cell sensitivity to standard chemotherapy drugs.

The authors investigated whether this phenotype can be rescued and show that (i) increased C/EBP ϵ expression is observed in dKO surviving cells, and (ii) exogenous expression of C/EBP α p30, C/EBP β LAP* and, partially, C/EBP β LAP isoform is able to rescue the growth defect in double KO mice.

To gain further insights into the nature of the defect, they conduct gene expression and chromatin accessibility assays and show a large number of changes in dKO cells, which can be partially rescued by C/EBP α p30 and C/EBP β LAP* overexpression.

In summary, the authors suggest a prominent role for different members of the C/EBP transcription factor family in the maintenance of MLL-ENL induced leukaemia. These results add an important piece to our understanding of MLL AML biology. The authors suggest that affected patients could be treated by interfering with non-oncogenic C/EBP addiction.

However, the manuscript as it stands cannot be published in its present form as outlined below. It needs to be seriously revised or even resubmitted after discussion with the editor.

In general, the paper is not well written, has some serious flaws and is extremely sloppy regarding the data presentation. It would benefit from drastic shortening, revision of the figures, removal of excessive speculation and also from editing by a native English speaker or maybe the senior author. It also lacks proper statistics and it is often not clear how often the experiments were done. I have seriously struggled with evaluating this paper, as it is really very time consuming to have to deal with trivia.

Figures 1 and S1.

The experiments showing that removing C/EBP β slows the growth of established MLL-ENL-transformed cells are believable. However, the figures need a better annotation, they need to be self-explanatory. The colony size distribution for WTFL and Cebpb-KO cells is shown at the bottom of Figure 1B. However, whilst we can guess, neither in the figure nor in the legend is clearly stated to which cell type each graph relates. Please add the scale to the pictures at the top of the same figure.

Figure 2: In this section, the authors show that simultaneous KO of Cebpa and Cebpb lead to increased sensitivity to drug treatment. I assume it is included to try and convey the idea that interference with C/EBP activity could be a therapeutic possibility. It is not discussed what such a therapeutic interference could be and how this would affect normal cells as they would be unable to differentiate similar to what is seen in CEBPA mutant AML. Moreover, this section is extremely weak as the differences between the already quite damaged control cells (even without deletion) and the drug treated cells are highly variable between the two experiments. Moreover, these experiments do not really add anything to the story. Frankly, I would recommend to take them out.

Figure 3:

Major issues: Here the authors try to show that Cebpe compensates for the lack of Cebpa/b. This claim is simply unsubstantiated. While Cebpe is clearly upregulated in dKO cells, it does by no means show that there is compensation. It is likely, but currently nothing more than a correlation. The authors say that they were unable to knock-down the gene, which is indicative of compensation, but there are other ways - knock-down C/EBPa/b in leukemic cells followed by a ChIP for C/EBPe - for example or show that Epsilon can rescue as well followed by CHIP (can be done with a tagged protein) and binds to the same sites. A significant overlap would enforce the hypothesis that C/EBPε overexpression can functionally compensate. I would therefore suggest to either do these experiments or tone down these statements everywhere.

Minor issues:

(i) The bar plot in figure 3A should be better labelled (for example, changing vertical axis label in "mRNA fold change expression") and a colour legend should be added to specify which cell types the data belong to. Moreover, in the figure legend, the experiment has been incorrectly reported as RNA-seq in spite of being RT-qPCR.

(ii) I am seriously puzzled by the complete absence of C/EBPe in Fig 2B WT cells although the RNA is present at the same level as that of the other family members. Why is that?

(iii) In the blot in figure 4a, the lane representing the dKO cells complemented with the p30 isoform shows two bands. The upper band molecular weight looks compatible with p42 expression. Similarly, the blot representing the dKO cells complemented with the single C/EBPβ isoforms show multiple bands, compatible with the simultaneous expression of more than one isoform. However, as the experiment was conducted on Cebpa/Cebpb dKO cells using an anti-FLAG antibody, it should be observed only the exogenous isoform expression. How do the authors explain that? Also, the experiment is incorrectly reported as anti-FLAG immunoprecipitation.

Figure 4: The rescue experiments showing that some C/EBPa and B proteins can rescue dKO cell growth are believable although a bit of better explained statistics in all of the experiments would make it even more so. How often were these experiments done? If this is a one-off, it cannot be published and therefore this information should be included in the figure legends.

Minor issues: (i) Again, the figures are poorly labelled. Whilst the colour code is shown in 4A, it is by no means a given that it applies also to the rest.

(ii) On page 8, line 181 is stated that "the C/EBPb LAP* isoform and, in part, the C/EBPa LAP or p30 isoforms further restored dKO MLL-ENL- transformed cell proliferation". The authors confused the isoform names.

(iii) How do the authors explain the sudden decrease in WT cell proliferation at the end of the curve in fig.4F?

Figure 5:

Major issues: (i) This section is utterly confusing, badly presented and requires a complete overhaul. The lettering is way too small. All other Figures are generously proportioned, why is this one so cluttered? Some of the panels (5 I, J, K) should go into the supplements.

(ii) The authors defined the gene expression profile for two clones, referred to as "clone 1" and "clone 2". It is not clear if these clones are derived from different mice or they are the same clones 1 and 2 previously showed in fig. 3B (i.e. derived from the same mouse). This is important, because the overlap of genes commonly up and downregulated in the two dKO clones compared to control (p8 line 192-193, Venn diagram in figure 5A) is very low and its significance could be questioned. In the main text - but not in the figure legend - it says that the RNA-Seq data are from triplicates - independent isolates? Clarify. In addition: what are these genes? That would make this analysis at least a good resource.

(iii) In the same vein, the heatmaps in figure 5E and F relative to gene expression seem to depict some substantial differences among triplicates. I would suggest the authors to assess the extent of

intra and inter group variability (i.e. by principal component analysis (PCA) and Pearson's coefficient calculation) to test the quality of the data.

Other issues:

- (i) In figure 5D a colour key with the value scale is missing and it is not even stated in the figure legend what is what.
- (ii) On page 10, line 222-224 it is stated that "Specifically, *Csf1r*, *Rel* family members (*Rel*, *Rela*, *Relb*), and *Tgfb2* levels were decreased in the dKO cells, supporting the pioneering role of *C/EBP α* and *C/EBP β* in myeloid commitment". The authors should add appropriate references to justify this statement.
- (iii) On page 10, line 229-231 it is stated that "Consistent with the reduced myeloid commitment observed in the dKO cells, we detected strong dysregulation of the inflammatory genes, documented by significant enrichment of "hallmark inflammatory response" and "hallmark *Tnfa* signaling via *Nfkb*" ". Again, the authors should add appropriate references to justify this statement.
- (iv) Figure 5E, indicate what the colour key represents. Logs? Not logs?
- (v) On page 11, line 248-250, the sentence "Focusing on genes that exhibited both, re-establishing gene expression and restoring the chromatin status in dKO-LAP* cells to WTFL status yielded a list of genes, part of which is shown in Figure 5F" is not very clear. Please, rephrase.
- (vi) In the barely legible figure 5F there are several (I assume mislabelling) errors. The heat map relative to RNA-seq data shows that *Cebpb* expression decreases in the dKO and then increases after LAP+ expression. This is completely incompatible with a *Cebpb* KO background. At the same way, the ATAC-seq heatmap reports three genes labelled as *Cebpb*. In the same figure, as well in the figure legend, is not specified what the colour key indicates.
- (vii) It is not specified (neither in the text nor in the figure or the figure legend) whether the motif search showed in figure 5G was performed in dKO-LAP* specific peaks. I would also suggest showing the enrichment p-value, together with the number of target sequences containing the indicated motifs, for all the three genotypes, otherwise it is not possible to judge whether there is any enrichment for the indicated binding sites in dKO-LAP* cells.
- (viii) In figure 5H should be stated what each genome browser track represents (i.e. ChIP or ATAC-seq). The authors should also mention in the main text which ChIP data are reported and why.
- (viiii) In figure 5I top panel, the authors show that *Igf1* is expressed in the WTFL and dKO-LAP* cells, but not in the dKO cells, by comparing RNA-seq reads. However, read counts are strongly affected by the total number of reads obtained during sequencing, a parameter that can differ a lot among libraries. Thus, would be more appropriate to show the data in term of normalized read counts, i.e. RPKM. It would also be better to condensate the triplicates in one bar, showing appropriate statistics.
- (x) In figure 5I bottom panel, statistics should be added. Again, how often were these experiments done?
- (xi) In figure 5J p-values are shown, even if not significant. On the other hands, in figure 5K non-significant p-value are not reported. Please, unify statistics presentations.

Discussion

In general, the discussion is way too long, repetitive - the whole first paragraph mirrors the introduction - and contains way too much speculation given the limited message. See my remarks above re compensation by *C/EBP ϵ* and *C/EBP* therapy. The paragraph about *SWI/SNF* is also completely confusing in the absence of an indication in Figure 4 A which isoform of *C/EBP β* associates with it and why we are even looking at the different isoforms.

- (i) Pag12 line 295, Chen et al., 2000 is not an appropriate reference. In this paper the authors show that *Cebpb* can functionally replace *Cebpa* in liver but not in adipose tissue when expressed at the *Cebpa* locus. Hematopoiesis is not mentioned at all. Please tone down or provide another reference.

(ii) On page 13, line 308-309, the authors raised the possibility that the rare event of C/EBP ϵ expression in their MLL-ENL-transformed model may have occurred before C/EBP α or C/EBP β deletion. If this was true, after prolonged WTFL cells culture, an increase in C/EBP ϵ expression should be expected. However, this does not happen, as showed in supplementary figure 3.

Methods: In general, the method section should be more detailed, and contain more information about how data have been analysed. In particular, a better description of the C/EPB isoform constructs used for complementation experiments is needed, information about anti-Flag antibody used for immunoblotting is missing, the RNA-seq library preparation description is missing. Sequencing data analysis should be described more in detail (for example, parameters used with bioinformatics software, data normalization strategy, etc.)

Data availability: Raw sequencing data should be available. A table containing RPKM, fold change and p-value relative to all sequencing analysis carried on should be added.

Point-by-point responses to the reviewers' comments:

We would like to thank the reviewers for their critical insightful comments and suggestions that have helped to improve and further strengthened our manuscript.

Reviewer #1 (Comments to the Authors (Required)):

The manuscript by Wesolowski et. al. studies the potential role for *cebpb* and *cebpe* in the maintenance of MLL-ENL driven leukemia, particularly in the absence of *cebpa*. This line of studies is prompted by previous observations that *cebpa* is required for initiation, but not maintenance of leukemia. They establish a convincing role for *cebpb* as relevant in this setting and a suggestive role for *cebpe*. This set of studies provides important understanding of MLL-ENL biology and the role of *ceb* proteins in the setting of myeloid progenitor transformation. However, there are specific aspects that could be clarified to enhance impact.

We thank the reviewer for pointing out the impact of our data.

Comments:

1. The authors mention the inability to suppress *cebpe* expression via shRNA or siRNA. This problem is understandable in this system. However, many have been able to inactivate genes by CRISPR mediated approaches in these cells. This would be a very helpful experiment in terms of validating the role of *cebpe* if it could be completed.

Only 14 cell clones were retrieved from 4 Million transformed cells (following deletion of C/EBP α,β ; frequency: less than 1 per 250 000 cells; termed: C/EBP α,β dKO C/EBP ϵ^+ to indicate expression of resident C/EBP ϵ). These data suggest that only C/EBP ϵ expressing cells survive C/EBP α,β deletion. The C/EBP α,β dKO C/EBP ϵ^+ cells are prone to apoptosis and difficult to handle in tissue culture, suggesting that C/EBP ϵ is an inferior replacement for C/EBP α,β . Several carefully controlled approaches to knock down C/EBP ϵ by RNA interference failed, probably because of the delicate condition of the C/EBP α,β dKO C/EBP ϵ^+ cells and because of the general problem to inactivate essential components and proving direct causal relationship (C/EBP ϵ removal after removal of C/EBP α,β).

Nevertheless, we do see that the manuscript would substantially benefit from an approach to remove C/EBP ϵ in dKO cells (reviewer#1 and #3). Accordingly, we have made additional efforts to show that C/EBP ϵ is essential in dKO by combining genome editing of the resident C/EBP ϵ gene with genetic complementation by C/EBP β LAP (which showed the best complementation in all experiments, as e.g. shown in Figure 4). The data (new Figure 5; line 212 to 258 in the manuscript) show that C/EBP ϵ can be deleted/inactivated only in dKO cells that ectopically express C/EBP β LAP* but not in parental dKO cells. Interestingly, parental C/EBP ϵ dKO cells permit occasional occurrence of in frame C/EBP ϵ deletions that all affect previously identified negative regulatory regions in the C/EBP ϵ protein, suggesting that C/EBP ϵ activation but not its removal was tolerated in C/EBP ϵ dKO cells.*

2. It is hard to tell in the manuscript if they ever assessed leukemia generation as opposed to in vitro cell proliferation/survival of the various genotypes. If not, this does influence the interpretation of some of the data and thus needs to be clearly pointed out in the discussion. Some of the results regarding dependency might be different if growth were assessed in vivo. This caveat should be clearly stated.

We now state in the discussion (first paragraph, last sentence) that data presented in this manuscript are based on MLL-transformation in vitro.

3. The cells transformed here were from fetal liver whereas most published experiments assess HSC/progenitors from postnatal mice. This could influence results since these cell types are quite different. This should be similarly pointed out in the discussion.

Indeed, most experiments involving MLL fusion proteins have been done with bone marrow derived HSPCs. On the other hand, MLL rearranged leukemia is predominantly seen in infants and children and there are many indications that the true target of transformation may represent an embryonal (fetal liver?) hematopoietic precursor. Hence, it has been argued that fetal liver cells would be even the more appropriate target (Sinha et al. 2020, Exp. Hematol. 85:13-19). In addition, human adult bone marrow HSC are less efficiently immortalized by MLL-fusions in comparison to fetal liver cells, even after enriching for the CD34+ fraction (Horton et al., 2013, Leukemia 27:1116–1126). At present there is no good scientific evidence that would favor one cell type over the other. In addition, there is no precedence that basic principles of transformation would be different in fetal liver vs. bone marrow. Therefore, we feel that a fetal liver based MLL-ENL model yields valid data.

4. The authors seems to propose that cebp proteins and Hox proteins are working together to maintain a leukemia gene expression program. This is reasonable and in line with previous publications. Figure 5C is particularly important. Where did this gene set come from? There should be increased explanation of this figure as much of the argument around the function of cebp relies on this data. This is particularly true given the modest change in Hox gene expression which is nicely pointed out by the authors.

The gene set used was derived from Collins, et al. and Jay L. Hess, PNAS 2014. The text has been slightly changed to name gene set origin and regulation observed in our experiments (line 292-299).

Reviewer #2 (Comments to the Authors (Required)):

The paper of Radoslaw Wesolowski, et al. reported that removing C/EBPb alone or both C/EBPa and C/EBPb slowed the growth and reduced the viability of MLL-ENL-transformed cells; C/EBPe showed increased expression in dKO-MLL-ENL colonies; furthermore ectopic expression of p30 and C/EBPb in C/EBPa/b KO cells partially rescued MLL-ENL-transformed cell growth. Thus the authors concluded that myelomonocytic MLL transformation and its maintenance remains continuously dependent on C/EBP family members.

Suggestions for improvement: major

1) It has been reported by multiple groups that C/EBPa is not required by the maintenance of MLL-ENL transformed cells. In this paper the authors tested the hypothesis that other C/EBP family members can contribute to the maintenance of the leukemic phenotype. In figure 1, the authors examined the role of C/EBPb in leukemic phenotype maintenance and showed loss of C/EBPb did not affect the number of colonies but reduced cell growth and viability. To better understand the role of C/EBPb in MLL-ENL transformation maintenance and to compare C/EBPa:C/EBPb Dko-MLL-ENL cells, it is suggested that the properties of these C/EBPb KO MLL-ENL cells could be further characterized, including colony formation in a serial replating assay (which correlates with in vivo leukemogenesis), cell cycle, apoptosis, and the expression of other C/EBP family members.

Addressing suggestions of reviewer #2 and #3:

We included experimental data (new Figure 2 and text lines 139 to 175), comparing the phenotypes of MLL-ENL WT^{FL}, C/EBPb KO and C/EBP α,β dKO C/EBP ϵ^+ cells using flow cytometric analysis (cell surface markers, including Ly6C, Ly6G, CD11b, CD117, CD115), determination of apoptosis rate (Annexin V/7-AAD), proliferation frequency (CFSE dye dilution; Figure 2B), cytochemical staining of cytopins (Figure 2C), serial replating (Figure 2D,E) and cytokine dependency (WT vs cebpb KO, Figure 3F). Phenotypic differences between genotypes are now apparent.

2) In Figure 2a the authors compared the response of WT-MLL-ENL cells vs C/EBP α :C/EBPb dKO-MLL-ENL cells to the chemotherapy drug cytarabine, daunorubicin, or both. The authors concluded that dKO-MLL-ENL cells showed more apoptosis and more sensitivity to chemotherapy. However, it is not clear the relevance of this data to the main hypothesis of the paper. Furthermore, although the authors reported dKO-MLL-ENL were more sensitive to chemotherapy, if one looks at the percentage of annexin V+ following cytarabine treatment, the percentage of apoptotic/dead cells in WT MLL-ENL transformed cells increased from 38.7% to 63.5% (1.6 fold) in clone 1 and 16.37% to 64.7% (4 fold) in clone 2. In dKO MLL-ENL cells, the percentage of annexin V cells increased from 48.4% to 82.4% (1.7 fold) in clone 1 and from 45.5% to 85% (1.9 fold) in clone 2 (Figure 2a). Similar results were observed with other treatments. Based on these results, it is hard to conclude that dKO MLL-ENL cells are more sensitive. Mechanistically, cytarabine and daunorubine are targeting proliferating cells, while loss of C/EBPb slowed the proliferation, so it is hard to explain why dKO MLL-ENL should be more sensitive to these chemotherapeutic drugs. Furthermore, it is hard to assess responsiveness to chemotherapy without a good non-leukemic control, it would have been useful to have utilized nontransformed GMP as a control. Again, it is not clear how much this adds to the manuscript.

According to the reviewer's comments (reviewer #2 and reviewer #3, see below), the previous Figure 2A (and text within the results and discussion sections) has been removed from the manuscript.

3) Although the authors reported that removing both C/EBP α and C/EBPb reduced the number of viable MLL-ENL transformed cells (line 118-135), a thorough characterization of C/EBP α :C/EBPb dKO-MLL-ENL cells are necessary. A figure with side by side comparison between C/EBPb KO and dKO leukemic cells would be helpful. These analysis should include serial replating, cell cycle, and apoptosis.

(see also point #1 of reviewer #2; above) We included side by side comparisons, as requested by the reviewer (new Figure 2 and text line 139 to 175) in the revised version.

4) The data in Figure 5B data suggested very mild changes in expression of many MLL-ENL target genes following C/EBP α and C/EBPb deletion. It is also interesting to notice that C/EBP α and C/EBPb dKO did not affect the HSC enriched gene signature, an indicator for stemness. Given these results, one would argue that MLL-ENL alone might be sufficient to maintain LSC status, while C/EBP α , C/EBPb or C/EBP ϵ more likely contributes to leukemic cell proliferation, survival and myeloid identity. Could authors comment on this?

This is an interesting concept but difficult to disclose. Our data show that MLL transformed myeloid cells are addicted to CEBP and that C/EBP β LAP promotes proliferation and survival while compound removal of C/EBP α,β also affects clonogenicity. The RNA profiling data suggest a (potentially neomorphic) phenotype in C/EBP α,β cells that retains stemness and a shifted inflammatory state. Moreover, the new flow cytometry and cytospin data show that alteration of the genetic compositions of C/EBP family members modify their myeloid differentiation and that WT vs Cebpb KO clones have different cytokine requirements (Figure 2). In an effort to separate leukemic cell proliferation, survival, and myeloid identity one may try in the future e.g. to combine MLL-ENL e.g. with Bcl2 to overcome apoptosis and examine whether all C/EBP family member genes can be removed (hoping that Bcl2 would*

compensate for lack of C/EBP induced hematopoietic/myeloid growth factor/cytokine receptor expression and signaling pathways) and then examine phenotypes of depleted cells (if any cells will survive). However, a negative outcome of such type of experiments would be difficult to interpret and are currently beyond the scope of the manuscript.

5) Figure 5F showed a nice correlation between mRNA expression and chromatin conformation for a few of genes following C/EBP dKO and dKO-LAP rescue. How does this correlation apply to other up and down genes?

The full set of correlated genes is attached as a heatmap (Appendix #1), see end of this document).

Minor Suggestions

1) Line 133: "14 Cebpa/Cebpb dKO subclones from two mice (2 and 14 subclones, respectively)". Is (2 and 14) a typo? Should it be (2 and 12)?

2 and 12 is correct (originating from different animals) and has been corrected in the text and Figure 3.

2) Figure 4E, the color label did not match with the color labels in Figure 4A

The color has now been adjusted; we thank the reviewer for both comments.

Reviewer #3 (Comments to the Authors (Required)):

Summary:

Previous work demonstrated that C/EBP α is essential for MLL-ENL induced transformation, however, once transformation has taken place, its expression is not required for leukaemia maintenance. Wesolowski et colleagues explored the possibility that other members of the C/EBP transcription factor family are required to sustain the tumour once it is established.

The authors show that Cebpb or combined Cebpa/Cebpb gene deletion affects growth and viability of established leukemic cells in an MLL-ENL-transformed murine cell model. Simultaneous Cebpa/Cebpb knock-out (dKO) augments cell sensitivity to standard chemotherapy drugs.

The authors investigated whether this phenotype can be rescued and show that (i) increased C/EBP ϵ expression is observed in dKO surviving cells, and (ii) exogenous expression of C/EBP α p30, C/EBP β LAP* and, partially, C/EBP β LAP isoform is able to rescue the growth defect in double KO mice.

To gain further insights into the nature of the defect, they conduct gene expression and chromatin accessibility assays and show a large number of changes in dKO cells, which can be partially rescued by C/EBP α p30 and C/EBP β LAP* overexpression.

In summary, the authors suggest a prominent role for different members of the C/EBP transcription factor family in the maintenance of MLL-ENL induced leukaemia. These results add an important piece to our understanding of MLL AML biology. The authors suggest that affected patients could be treated by interfering with non-oncogenic C/EBP addiction.

However, the manuscript as it stands cannot be published in its present form as outlined below. It needs to be seriously revised or even resubmitted after discussion with the editor.

In general, the paper is not well written, has some serious flaws and is extremely sloppy regarding the data presentation. It would benefit from drastic shortening, revision of the figures, removal of excessive speculation and also from editing by a native English speaker or maybe the senior author. It also lacks proper statistics and it is often not clear how often the experiments were done. I have seriously struggled with evaluating this paper, as it is really very time

consuming to have to deal with trivia.

We would like to thank the reviewer for his detailed criticism and valuable suggestions that helped to improve the manuscript.

Figures 1 and S1.

The experiments showing that removing C/EBP β slows the growth of established MLL-ENL-transformed cells are believable. However, the figures need a better annotation, they need to be self-explanatory. The colony size distribution for WTFL and Cebpb-KO cells is shown at the bottom of Figure 1B. However, whilst we can guess, neither in the figure nor in the legend is clearly stated to which cell type each graph relates. Please add the scale to the pictures at the top of the same figure.

Scale bars have been included and labels (lost during processing of the Figure) have been added.

Figure 2: In this section, the authors show that simultaneous KO of Cebpa and Cebpb lead to increased sensitivity to drug treatment. I assume it is included to try and convey the idea that interference with C/EBP activity could be a therapeutic possibility. It is not discussed what such a therapeutic interference could be and how this would affect normal cells as they would be unable to differentiate similar to what is seen in CEBPA mutant AML. Moreover, this section is extremely weak as the differences between the already quite damaged control cells (even without deletion) and the drug treated cells are highly variable between the two experiments. Moreover, these experiments do not really add anything to the story. Frankly, I would recommend to take them out.

We removed the data and text according to suggestions of reviewer #2, point 2 and reviewer#3.

Figure 3:

Major issues: Here the authors try to show that Cebpe compensates for the lack of Cebpa/b. This claim is simply unsubstantiated. While Cebpe is clearly upregulated in dKO cells, it does by no means show that there is compensation. It is likely, but currently nothing more than a correlation. The authors say that they were unable to knock-down the gene, which is indicative of compensation, but there are other ways - knock-down C/EBP α /b in leukemic cells followed by a CHIP for C/EBP ϵ - for example or show that Epsilon can rescue as well followed by CHIP (can be done with a tagged protein) and binds to the same sites. A significant overlap would enforce the hypothesis that C/EBP ϵ overexpression can functionally compensate. I would therefore suggest to either do these experiments or tone down these statements everywhere.

(also see answer to reviewer #1, point 1) According to the comments/suggestions of the reviewers we included data obtained by Cas9 mediated gene editing of resident Cebpe. Cebpe could only be removed when C/EBP α,β dKO C/EBP ϵ^+ cells expressed LAP ectopically (Figure 5C – G), but not in dKO control cells. The new data are based on somatic genetics/gene editing and fully support our previous conclusion about the surrogate essential function of Cebpe for survival of the Cebpa/Cebpb dKO MLL-ENL transformed myeloblasts.*

Minor issues:

(i) The bar plot in figure 3A should be better labelled (for example, changing vertical axis label in "mRNA fold change expression") and a colour legend should be added to specify which cell types the data belong to. Moreover, in the figure legend, the experiment has been incorrectly reported as RNA-seq in spite of being RT-qPCR.

We apologize for the confusion that occurred due to the data presentation of our previous Figure 3A. In the previous version we took the normalized RNA-seq data and calculated the fold change of Cebp family members against the respective expression in WT cells (set to 1

for every Cebp-family member). The previous display was chosen to emphasize the enhancement of Cebpe expression in the dKO cells relative to WT and all other family members, but, as pointed out by the reviewer, eliminated the information of Cebp-family member expression in relation to each other.

As suggested by the reviewer, we corrected the misleading representation and now just show the normalized RNA-seq read counts. Accordingly, the vertical axis labeling in Figure 3A has been corrected to “normalized RNAseq reads” and both panels have been marked in accordance to the protein blots, as shown below in Figure 3B. The corresponding Figure legend has been corrected.

(ii) I am seriously puzzled by the complete absence of C/EBPε in Fig 2B WT cells although the RNA is present at the same level as that of the other family members. Why is that?

(assuming the reviewer meant Figure 3B and not 2B) In the previous Figure 3A we individually calculated the expression fold change of Cebp family members to the expression in WT^{FL} (set to 1; see point before) to demonstrate enhanced expression of Cebpe in dKO over WT^{FL} (accordingly, there are no red bars for Cebpa/b in the dKO, similar expression of Cebpd in the WT and dKO, while Cebpe expression is strongly increased). However, by showing now normalized RNA-seq read counts (and not fold change), it is possible to see that Cebpe is the weakest expressed Cebp family member in WT cells. Therefore, it is likely that its expression cannot be visualized by qualitative/semi-quantitative Western blotting. Blots are processed individually with different antibodies and adjusted to similar intensity of bands between data sets. Variables are antibody quality, avidity, secondary antibodies, exposure times and expression fluctuation. Thus, expression quantity can be estimated between the same antigens and only in relation to each other, but not between different antigens (on a strictly quantitative basis, such as molarity, copy numbers). Although the C/EBPε protein is obviously missing at comparable exposure times in WT^{FL} , prolonged overexposure (that leads to entirely burned out C/EBPε protein bands in the dKO samples), revealed some trace expression of C/EBPε also in the WT^{FL} . We feel that these issues are beyond the scope of the manuscript.

In this respect it was further recently reported that even though microglia express Cebpb mRNA, the protein abundance is regulated post-translationally by the ubiquitin ligase COPI and, as a consequence, C/EBPβ protein is barely detectable in these cells (Ndoja et al., 2020, Cell). Accordingly, protein expression data of C/EBPs are functionally more relevant and shown in Figure 3B. We do not know whether C/EBPε stability or translation or both are regulated, however, we have also identified upstream open reading frames in vertebrate C/EBPε that may regulate protein translation (unpublished data). Again, these issues are beyond the scope of the manuscript.

(iii) In the blot in figure 4a, the lane representing the dKO cells complemented with the p30 isoform shows two bands. The upper band molecular weight looks compatible with p42 expression. Similarly, the blot representing the dKO cells complemented with the single C/EBPβ isoforms show multiple bands, compatible with the simultaneous expression of more than one isoform. However, as the experiment was conducted on Cebpa/Cebpb dKO cells using an anti-FLAG antibody, it should be observed only the exogenous isoform expression. How do the authors explain that? Also, the experiment is incorrectly reported as anti-FLAG immunoprecipitation.

These are all valid comments that are resolved as follows:

C/EBPa blot and p30 lane: there is no resident C/EBPa / C/EBPβ detected in the dKO (also compare to Figure 3). The p30 CEBPa vector did not contain any p42 specific N-terminal sequence, accordingly, we can exclude that the band is p42 CEBPa. The upper band,

questioned by reviewer #3, is occasionally seen in conjunction with p30 in lysates and most likely represents sumoylated p30 CEBPa.

C/EBPβ blot: Multiple western bands in single C/EBPβ isoform complemented dKO cells are explained as follows: Both, LAP and LAP retained the internal (downstream) LIP start codon and may therefore show minute expression of the truncated LIP isoform. The band in the middle between LAP/LIP in the LAP-lane (lane #3) most likely represents a minor degradation product of full-length LAP. Minor bands directly above the major LAP*/LAP bands and band fuzziness of LAP*/LAP are due to multiple post-translational modifications which are abundant on both isoforms. Faint bands, further above at the top of the blots, are most likely sumoylated LAP*/LAP proteins.*

The protein blots in Figure 4 show lysate controls and have now been labelled correctly. We thank the reviewer for pointing this out.

Figure 4: The rescue experiments showing that some C/EBPa and B proteins can rescue dKO cell growth are believable although a bit of better explained statistics in all of the experiments would make it even more so. How often were these experiments done? If this is a one-off, it cannot be published and therefore this information should be included in the figure legends.

Side by side comparisons in different combinatorial settings using C/EBP isoforms were done 2-3 times with essentially similar results (except p42 expression, as data collected over many years and from numerous experimental settings repeatedly showed that p42 always resulted in growth inhibition). Figure 5 now also contains data on many clonal complementation assays with C/EBPβ LAP and genomic analysis of 31 clones. My lab never publishes one-time-only results.*

Minor issues: (i) Again, the figures are poorly labelled. Whilst the colour code is shown in 4A, it is by no means a given that it applies also to the rest.

It was stated before in legend to Figure 4A that the color code applies to B-G. We have now improved Figure 4 to display the color code for Figure 4B-G as an inset.

(ii) On page 8, line 181 is stated that "the C/EBPb LAP* isoform and, in part, the C/EBPa LAP or p30 isoforms further restored dKO MLL-ENL- transformed cell proliferation". The authors confused the isoform names.

We thank the reviewer for pointing this out [has been corrected; now lines 209-210: "In summary, the C/EBPβ LAP isoform and, in part, the C/EBPβ LAP or C/EBPa p30 isoforms restored dKO MLL-ENL transformed cell proliferation (Figure 4B, 4E) and viability (Figure 4C, 4F)."]*

(iii) How do the authors explain the sudden decrease in WT cell proliferation at the end of the curve in fig.4F?

We avoided disturbance of cultures that may have affected proliferation kinetics (e.g. by medium change that would have required a centrifugation step) and kept cells for 96h in the same medium/culture dish. The decrease in WT cell proliferation (Figure 4F) at the end of the growth curve is indicative of medium exhaustion at plateauing high cell density (also observed in Figure 1 C for WT^{FL} and Cebp KO).

Figure 5:

Major issues: (i) This section is utterly confusing, badly presented and requires a complete overhaul. The lettering is way too small. All other Figures are generously proportioned, why is this one so cluttered? Some of the panels (5 I, J,K) should go into the supplements.

We have improved the legibility of labels and reduced some of the data (genes & ATAC peaks) displayed in Figure 6E, F. All parts of Figure 6 (previously Figure 5) cover the issue of CEBP-deficiency and complementation on the level of molecular genetics and function. We think that the data sets belong together in one Figure and that the IGF1 data should be included and not shown as supplemental data, as IGF1 has been identified as a common MLL-ENL/Hox/Cebp target gene and its expression enhances growth/survival of transformed cells.

(ii) The authors defined the gene expression profile for two clones, referred to as "clone 1" and "clone 2". It is not clear if these clones are derived from different mice or they are the same clones 1 and 2 previously showed in fig. 3B (i.e. derived from the same mouse). This is important, because the overlap of genes commonly up and downregulated in the two dKO clones compared to control (p8 line 192-193, Venn diagram in figure 5A) is very low and its significance could be questioned. In the main text - but not in the figure legend - it says that the RNA-Seq data are from triplicates - independent isolates? Clarify. In addition: what are these genes? That would make this analysis at least a good resource.

Data from previous Figure 5 are now shown as overhauled version, Figure 6. Clones 1 (cl. #5) and clone 2 (cl. #13) are the same as in Figure 3 (where it was stated that they are derived from different mice).

The RNA-Seq data were generated from cell samples (independent experimental setups) harvested at different time points. Both overlaps of the up-regulated and down-regulated genes are significant ($p < 0.05$). According to the reviewer's suggestion the data (overlap up/down regulated) are now shown in a Supplementary Table 1.

(iii) In the same vein, the heatmaps in figure 5E and F relative to gene expression seem to depict some substantial differences among triplicates. I would suggest the authors to assess the extent of intra and inter group variability (i.e. by principal component analysis (PCA) and Pearson's coefficient calculation) to test the quality of the data.

Concerning the heatmaps in Figure 6E and 6F, we added PCA and sample correlation plots (information for reviewer: Appendix #2) showing the high correlation of the samples. We agree with the reviewer that the figure can be improved with respect to sample comparability. We therefore included only those genes of the two gene sets that are differentially expressed with an adjusted p -value of < 0.05 and a FC of at least 2. In addition, the display of the heatmap has been enlarged.

Other issues:

(i) In figure 5D a colour key with the value scale is missing and it is not even stated in the figure legend what is what.

A color key has been included in Figure 6D in the revised manuscript.

(ii) On page 10, line 222-224 it is stated that "Specifically, *Csf1r*, *Rel* family members (*Rel*, *Rela*, *Relb*), and *Tgfb2* levels were decreased in the dKO cells, supporting the pioneering role of *C/EBP α* and *C/EBP β* in myeloid commitment". The authors should add appropriate references to justify this statement.

Jaitin et al., 2016; Tamura et al., 2017; and Wang et al. 2016 references have been added.

(iii) On page 10, line 229-231 it is stated that "Consistent with the reduced myeloid commitment

observed in the dKO cells, we detected strong dysregulation of the inflammatory genes, documented by significant enrichment of "hallmark inflammatory response" and "hallmark Tnfa signaling via Nfkb" ". Again, the authors should add appropriate references to justify this statement.

These are terms from MSigDB, commonly used for gene set enrichment analysis, as mentioned in the legend to the figure. The statement is justified by our data, as shown in the figure and a reference to database and authors has been added in the text section (Liberzon et al. 2015); referring to MSigDB).

(iv) Figure 5E, indicate what the colour key represents. Logs? Not logs?

A color key has been added to Figure 6E shown as "log2".

(v) On page 11, line 248-250, the sentence "Focusing on genes that exhibited both, re-establishing gene expression and restoring the chromatin status in dKO-LAP* cells to WTFL status yielded a list of genes, part of which is shown in Figure 5F" is not very clear. Please, rephrase.

Text was rephrased (line 323-326):

"Focusing on genes that exhibited both, re-established gene expression and restored chromatin status in dKO-LAP cells towards the WT^{FL} status yielded a list of genes (highly correlated gene expression and chromatin status), part of which is shown in Figure 6F".*

(vi) In the barely legible figure 5F there are several (I assume mislabelling) errors. The heat map relative to RNA-seq data shows that Cebpb expression decreases in the dKO and then increases after LAP+ expression. This is completely incompatible with a Cebpb KO background. At the same way, the ATAC-seq heatmap reports three genes labelled as Cebpb. In the same figure, as well in the figure legend, is not specified what the colour key indicates.

The heatmap (Figure 6F, formerly 5F) shows RNA-seq data and indicates genes (e.g. Gramd3, Sgce and Scd1) that are re-activated as compared to WT after LAP expression. The data showing Cebpb RNA expression in the dKO+LAP* is entirely compatible and confirmative because LAP* represents the extended isoform of C/EBPβ and is expected to be detected in dKO+LAP* cells. We are aware that "LAP*, LAP, LIP" is not the most fortunate nomenclature that is used in the C/EBP literature.*

The second heat map in Figure 6F (formerly 5F) refers to ATAC peaks (probing genome structure). Several ATAC peaks (selected for peak significance) may show up in the same locus, such as e.g. shown in previous Figure 5 with Cebpb (3x) or IGF1 (3x). To eliminate this potential source of confusion, we now included only one peak per gene in the heatmap. Also, we adapted our previously used genome version to improve comparison of ATAC and RNA data. Analysis is now based on mm10. This slightly changes p-values for the motif enrichment, yet not the set of significant motifs (as shown in Figure 6 G). Color keys for the mRNA seq data as well as for the ATAC peak data in Figure 6F have been added.

(vii) It is not specified (neither in the text nor in the figure or the figure legend) whether the motif search showed in figure 5G was performed in dKO-LAP* specific peaks. I would also suggest showing the enrichment p-value, together with the number of target sequences containing the indicated motifs, for all the three genotypes, otherwise it is not possible to judge whether there is any enrichment for the indicated binding sites in dKO-LAP* cells.

Motif search was done on the full set of genomic regions defined by the ATAC peaks as described in Figure 6G, i.e. the peaks showing a correlation between RNA and ATAC data

(N=211). Number of target genes among the 211 targets and percentages in addition to p-values are now shown in Figure 6G.

(viii) In figure 5H should be stated what each genome browser track represents (i.e. ChIP or ATAC-seq). The authors should also mention in the main text which ChIP data are reported and why.

The genome browser tracks now have additional labels and the origin of the reported ChIP-seq data is mentioned by two references (Roe et al 2016; Zhong et al 2018) in the legend to Figure 6H. These data sets were chosen because they are of high quality.

(viii) In figure 5I top panel, the authors show that Igf1 is expressed in the WTFL and dKO-LAP* cells, but not in the dKO cells, by comparing RNA-seq reads. However, read counts are strongly affected by the total number of reads obtained during sequencing, a parameter that can differ a lot among libraries. Thus, would be more appropriate to show the data in term of normalized read counts, i.e. RPKM. It would also be better to condensate the triplicates in one bar, showing appropriate statistics.

Numerical p-values are now displayed in Figure 6 I-K and the RNAseq reads (Figure 6I) are now shown with statistics.

(x) In figure 5I bottom panel, statistics should be added. Again, how often were these experiments done?

Statistics has been added. The ELISA test was repeated 3 times.

(xi) In figure 5J p-values are shown, even if not significant. On the other hands, in figure 5K non-significant p-value are not reported. Please, unify statistics presentations.

We have unified the display of statistics in Figure 6 I-K. The issue of “significance” at the arbitrary threshold ≤ 0.05 remains a matter of discussion in life sciences. We realized that IGF1 production is low but it is functional, reproducible, observed by others and, importantly, in our context entirely consistent with independent evidence derived from gene expression, genome architecture, and responsiveness in relation to genotypes/-complementation.

Discussion

In general, the discussion is way too long, repetitive - the whole first paragraph mirrors the introduction - and contains way too much speculation given the limited message. See my remarks above re compensation by C/EBP α and C/EBP β therapy. The paragraph about SWI/SNF is also completely confusing in the absence of an indication in Figure 4 A which isoform of C/EBP β associates with it and why we are even looking at the different isoforms.

The discussion section has been shortened and partially re-ordered. The first paragraph indicating major results and context has been shortened to two sentences. Discussion about SWI/SNF was removed. All text relating to previous Figure 2 (chemotherapeutic connection) has been removed.

(i) Pag12 line 295, Chen et al., 2000 is not an appropriate reference. In this paper the authors show that Cebpb can functionally replace Cebpa in liver but not in adipose tissue when expressed at the Cebpa locus. Hematopoiesis is not mentioned at all. Please tone down or provide another reference.

The Chen et al. 2000 reference has been exchanged against “Jones et al. 2002, Blood”

presenting results that suggest “Expression of C/EBP β from the Cebpa gene locus is sufficient for normal hematopoiesis in vivo”.

(ii) On page 13, line 308-309, the authors raised the possibility that the rare event of C/EBP ϵ expression in their MLL-ENL-transformed model may have occurred before C/EBP α or C/EBP β deletion. If this was true, after prolonged WTFL cells culture, an increase in C/EBP ϵ expression should be expected. However, this does not happen, as showed in supplementary figure 3.

Even after prolonged cultivation of the MLL-ENL transformed WT^{FL} cells we did not find upregulation of C/EBP ϵ , however, we do not know whether few C/EBP ϵ expressing cells that may have been initially among the MLL-ENL transformed progenitors are diminished by overgrowth and surfaced only after elimination of Cebpa/Cebpb genes by positive selection. On the other hand, the observation of epigenetic downregulation of C/EBP ϵ after extended growth of CEBP β LAP expressing dKO cells suggests crossregulation of Cebpe gene expression (which might occur on the gene expression or protein expression level or both, - has not been investigated). We deduced from the unfavorable growth properties of dKO C/EBP ϵ ⁺ cells that C/EBP ϵ upregulation is the last option for transformed cell survival once Cebpa/Cebpb genes have been eliminated.*

Methods: In general, the method section should be more detailed, and contain more information about how data have been analysed. In particular, a better description of the C/EPB isoform constructs used for complementation experiments is needed, information about anti-Flag antibody used for immunoblotting is missing, the RNA-seq library preparation description is missing.

Additional information has been added to several parts of the Methods section. Information about the C/EBP β isoform constructs had been described previously (Kowenz-Leutz et al 1994; Stoilova et al 2013, Cirovic et al. 2017) and is cited in the reference section. Antibody information has been added to the Materials section (lines 670-673). RNAseq information has been added and information on cytofluorometric analysis and serial replating (new Figure 2) has been added.

Sequencing data analysis should be described more in detail (for example, parameters used with bioinformatics software, data normalization strategy, etc.)
Data availability: Raw sequencing data should be available. A table containing RPKM, fold change and p-value relative to all sequencing analysis carried on should be added.

Raw and processed data have been deposited in GEO and are available under the accession ID GSE153624. More detail has been added to the sequencing data analysis section.

Appendix #1

Appendix 1

October 19, 2020

RE: Life Science Alliance Manuscript #LSA-2020-00709-TR

Prof. Achim Leutz
Max Delbrück Center for Molecular Medicine
Tumorigenesis and Cell Differentiation
Robert-Rössle-Strasse 10
Berlin, Berlin 13125
Germany

Dear Dr. Leutz,

Thank you for submitting your revised manuscript entitled "Myeloid Transformation by MLL-ENL Depends Strictly on C/EBP". We would be happy to publish your paper in Life Science Alliance pending final revisions in accordance to the reviewer 1's request and necessary to meet our formatting guidelines.

Along with the points listed at the end of this email, please also attend to the following:

- please provide your manuscript text in editable doc format
- please make sure that the manuscript sections are ordered in accordance to LSA's formatting guidelines (<https://www.life-science-alliance.org/manuscript-prep#format>)
- please submit the RNA-Seq and ATAC-Seq data in a publicly available database and provide the accession information in a separate Data Availability section (<https://www.life-science-alliance.org/manuscript-prep#datadepot>)
- please add your supplementary figure legends to the main manuscript text (directly under the main figure legends)
- please include supplementary table 1 and respective table legend in the manuscript. Please also add a callout to the Table S1 in the manuscript text
- please add a callout for Fig. 5E and Fig. S1A in your main manuscript text
- for figure S2, please change the subpanel labels from (A*, B*) to panels C&D

A. FINAL FILES:

B. MANUSCRIPT ORGANIZATION AND FORMATTING:

Sincerely,

Shachi Bhatt, Ph.D.

Executive Editor
Life Science Alliance
<https://www.lsjournal.org/>
Tweet @SciBhatt @LSAJournal

Reviewer #1 (Comments to the Authors (Required)):

The revised version is improved. However the authors argue there is no known difference in transformation based on cell of origin in response to one of my previous comments. This is incorrect. No new experiments are needed, but a statement in the conclusion that the cells of origin here are fetal liver (which may be more relevant for childhood leukemia) and that it is not clear if these exact dependency profiles would be the same in leukemias derived from postnatal HSPC would be prudent. This will help with clarity in case others do this experiment from adult HSC and get different answers.

Reviewer #2 (Comments to the Authors (Required)):

I believe the authors have addressed most of the comments of the previous review, and explained their efforts and/or why it might be difficult to perform every suggested experiment. I am satisfied that the paper is much improved and can be published in present form.

Reviewer #3 (Comments to the Authors (Required)):

The authors have gone to great length to address our many comments and the manuscript has been significantly improved and reads much better. Most importantly, they added more experiments to clarify the role of C/EBP ϵ . I am happy to see it published.

October 23, 2020

RE: Life Science Alliance Manuscript #LSA-2020-00709-TRR

Prof. Achim Leutz
Max Delbrück Center for Molecular Medicine
Tumorigenesis and Cell Differentiation
Robert-Rössle-Strasse 10
Berlin, Berlin 13125
Germany

Dear Dr. Leutz,

Thank you for submitting your Research Article entitled "Myeloid Transformation by MLL-ENL Depends Strictly on C/EBP". It is a pleasure to let you know that your manuscript is now accepted for publication in Life Science Alliance. Congratulations on this interesting work.

DISTRIBUTION OF MATERIALS:

Again, congratulations on a very nice paper. I hope you found the review process to be constructive and are pleased with how the manuscript was handled editorially. We look forward to future exciting submissions from your lab.

Sincerely,

Shachi Bhatt, Ph.D.

Executive Editor

Life Science Alliance

<https://www.lsjournal.org/>
